# Interplay of structure and photophysics of individualized rod-shaped graphene quantum dots with up to 132 $sp^2$ carbon atoms

Daniel Medina-Lopez[1,6], Thomas Liu[2,6], Silvio Osella[3], Hugo Levy-Falk [2], Nicolas Rolland[4], Christine Elias[2], Gaspard Huber [5], Pranav Ticku[2], Loïc Rondin [2], Bruno Jousselme[1], David Beljonne [4], Jean-Sébastien Lauret [2] ✉ & Stephane Campidelli [1] ✉

Nanographene materials are promising building blocks for the growing field of low-dimensional materials for optics, electronics and biophotonics applications. In particular, bottom-up synthesized 0D graphene quantum dots show great potential as single quantum emitters. To fully exploit their exciting properties, the graphene quantum dots must be of high purity; the key parameter for efficient purification being the solubility of the starting materials. Here, we report the synthesis of a family of highly soluble and easily processable rod-shaped graphene quantum dots with fluorescence quantum yields up to 94%. This is uncommon for a red emission. The high solubility is directly related to the design of the structure, allowing for an accurate description of the photophysical properties of the graphene quantum dots both in solution and at the single molecule level. These photophysical properties were fully predicted by quantum-chemical calculations.

Since the discovery of graphene, great attention has been devoted to the design and synthesis of nano-size graphene materials such as graphene quantum dots[1–4] (GQDs) and graphene nanoribbons (GNRs)[5–8]. Their large delocalized π-electron systems confer interesting and tunable properties such as a sizeable optical gap and electronic conductivity. To fully exploit the properties arising from their structure, nanographenes have to be synthesized by a bottom-up approach, leading to atomically precise objects.

For the last decade, a renewed interest has been dedicated to the synthesis of large polycyclic aromatic hydrocarbons (PAH) with controlled shapes, edges (zigzag *vs.* armchair) and functional groups. This has led to significant progress in the bottom-up synthesis of GQDs[9–14],

and in their theoretical descriptions[15–18]. However, the study of the optical properties of GQDs is still challenging because of the limited solubility of large PAH materials. While the photophysics of hexabenzocoronene derivatives is relatively well-known[19,20], almost no advanced optical characterization has been performed on larger GQD structures with ≈100 carbon atoms and above. Recently, we reported on single molecule spectroscopy experiments on bottom-up triangular-shaped GQDs containing 96 $sp^2$ carbon atoms. We showed that the GQDs are efficient and stable single-photon emitters at room and cryogenic temperatures[21–23]. To fully exploit the potential of these objects for quantum technologies[24], it is critical to address in depth their structure-properties relationship. This understanding would lead

[1]Université Paris-Saclay, CEA, CNRS, NIMBE, LICSEN, 91191 Gif-sur-Yvette, France. [2]Université Paris-Saclay, CNRS, ENS Paris-Saclay, CentraleSupélec, LuMIn, 91400 Orsay, France. [3]Chemical and Biological Systems Simulation Lab, Centre of New Technologies, University of Warsaw, Banacha 2C, 02-097 Warsaw, Poland. [4]Laboratory for Chemistry of Novel Materials, University of Mons, 7000 Mons, Belgium. [5]Université Paris-Saclay, CEA, CNRS, NIMBE, LSDRM, 91191 Gif-sur-Yvette, France. [6]These authors contributed equally: Daniel Medina-Lopez, Thomas Liu. ✉e-mail: lauret@ens-paris-saclay.fr; stephane.campidelli@cea.fr

to the possibility of developing a reverse engineering approach to design GQDs with tailor-made properties. To reach such a degree of control, some fundamental issues need to be overcome, the main one being the poor solubility of GQDs due to strong π-stacking interactions.

Firstly, the poor solubility leads to the formation of multimeric species in solution. The resulting optical properties are then modified by the aggregation state, giving rise to broader and poorly defined absorption and emission spectra[25–27]. Secondly, the low solubility prevents using standard purification methods of organic chemistry, leading to the presence of side products that complicates the analysis and the use of GQDs[22]. Therefore, the exploitation of the full potential of GQDs requires them to be truly individualized in solution and in the solid state.

To improve the solubility and decrease the stacking, two main strategies have been pursued in the literature: the functionalization of the periphery with bulky groups such as chlorine atoms, *tert*-butyl or mesityl units[27–33] and the formation of non-planar systems like propeller-type or twisted GQDs[34–44]. The strategy based only on the incorporation of bulky groups allowed for improving the apparent solubility of the GQDs but did not result in a real individualization of the molecules in solution (see discussion below on triangular-shaped $C_{96}$ containing *t*Bu and Supplementary Fig. 1). The non-planar approach led to outstanding graphene nanostructures exhibiting quasi-3D morphologies and chiroptical properties. However, the distortion of the structure may affect the effective π-conjugated systems[36,45]. It is worth mentioning that the morphology of twisted materials is difficult to predict since 5-, 7-, or 8-member rings can be created during the Scholl oxidation of hindered structures.

In this paper, we report on the synthesis of rod-shaped GQDs containing 78, 96, 114, and 132 $sp^2$ carbon atoms, respectively. The high solubility of the GQDs is obtained thanks to the specific location of bulky *tert*-butyl groups in their structure. It permits both the purification of the products and an in-depth study of the structure-property relationships through optical absorption spectroscopy, polarization-resolved and time-resolved photoluminescence (PL) spectroscopies in solution, and at the single molecule level. The comparison between the experimental results and theoretical predictions based on Density Functional Theory (DFT) and its time-dependent variant (TDDFT) provides a deep understanding of the optical properties of these GQDs. We show that the GQDs are highly individualized, and the excited states are fully delocalized on the whole structure with fluorescence quantum yields around or above 90%.

## Results

In our previous work, we had to overcome the high aggregation tendency of the triangular-shaped GQDs bearing dodecyl alkyl chains ($C_{96}$-$(C_{12}H_{25})_6$) to perform single molecule spectroscopy[21–23]. The alkyl chains and the aromatic core tend to self-organize giving rise to a columnar liquid crystal[46]. Here, we focused on using *tert*-butyl groups to improve the solubility and the individualization of the molecules in solution by suppressing the π-stacking interactions between the GQDs. This strategy was also used to synthesize twisted nanostructures but with the aim of preventing the formation of C-C bonds between adjacent phenyl rings, leading to non-planar systems[40,42]. The *tert*-butyl functional groups possess a number of assets: they are easy to incorporate into a structure, and their electron-donating character improves the efficiency of the Scholl reaction. Replacing alkyl chains with *tert*-butyl groups in triangular-shaped $C_{96}$ GQDs results in an apparent improvement in solubility accompanied by sharper absorption and emission bands (see Supplementary Fig. 1 for details). However, these GQDs still form aggregates (or at least dimers) as in the case of $C_{96}$ GQDs substituted with six mesityl groups[30]. Thus, we designed a $C_{96}$ rod-shaped GQD containing *tert*-butyl groups properly positioned on adjacent phenyl rings along the long axis of the GQDs (Fig. 1) to

prevent the π-stacking between molecules and permit the individualization of the molecules in solution.

## Synthesis of $C_{96}$-$t$Bu$_8$

The structure of $C_{96}$-$t$Bu$_8$ is depicted in Fig. 1a. Its synthesis is based on the Scholl oxidation of the corresponding polyphenylene dendrimer **1**. The full description of the synthesis is given in Fig. 6 (see "Methods") and in the Supplementary Discussion.

The dehydrogenation of the dendrimer is the key step for forming GQDs. Using $FeCl_3$ gives rise to chlorinated side products generated along with the desired nanographene particle. Because of the generally low solubility of GQDs, it can be a challenge to get rid of these chlorinated species, which is especially true as the nanographene particles increase in size[47]. The solubility of $C_{96}$-$t$Bu$_8$ GQD enables us to use classical purification techniques to obtain highly pure nanoparticles. Indeed, the GQDs were purified by size exclusion chromatography (SEC) on Bio-beads S-X1 or S-X3 using tetrahydrofuran (THF) as eluent. This separation is facilitated if, before the column separation, the nanoparticles are dispersed in THF and ultracentrifuged (see Supplementary Fig. 2 for details concerning the purification). The characterization of the $C_{96}$-$t$Bu$_8$ GQD before and after the purification step was performed using Matrix-Assisted Laser Desorption/Ionization-Time of Flight (MALDI-ToF) mass spectrometry (Supplementary Fig. 2b). It shows a clear increase of the purity of the sample. The $C_{96}$-$t$Bu$_8$ GQD was characterized by $^1$H-$^1$H and $^1$H-$^{13}$C 2D nuclear magnetic resonance (NMR) experiments and cyclic voltammetry (Supplementary Fig. 3).

## Photophysics of $C_{96}$-$t$Bu$_8$

The absorption and photoluminescence spectra of $C_{96}$-$t$Bu$_8$ in 1,2,4-trichlorobenzene are shown in Fig. 1b. In contrast with previous reports on such large GQDs, including ours, the absorption spectrum of $C_{96}$-$t$Bu$_8$ shows very sharp lines with almost no diffusion background. This is a first indication of the good individualization of the nanographene particles. The absorption spectrum comprises three main series of bands centered at 611 nm, 565 nm and 470 nm. Likewise, the PL spectrum shows two peaks at 622 nm and 678 nm and a weak band at ≈750 nm. The Stokes shift between absorption and emission is of 11 nm (≈36 meV). The TDDFT calculations, shown as black bars in Fig. 1b, match the experimental data with a bright lowest-lying singlet excitation centered at 615 nm (for 611 nm measured), and higher energy singlet states at 482 nm and 456 nm. We have optimized the geometry of $C_{96}$-$t$Bu$_8$ in the lowest singlet excited state $S_1$, using TDDFT, and have mapped the structural distortions between $S_0$ and $S_1$ onto the ground-state normal modes. We then applied a displaced, undistorted, harmonic oscillator model to calculate the spectral lineshape of the first optical transition in the Franck-Condon approximation (see Methods). The resulting absorption and emission spectra are depicted in Fig. 1d. Looking in more detail at the absorption spectra (calculated and measured), one can notice the presence of a vibrational progression associated with the main absorption bands. For instance, the calculated (respectively, the measured) lines centered at 566 nm (563 nm) and 558 nm (552 nm) correspond to the vibronic (0–1) replica of the 0-0 line at 615 nm (611 nm). Likewise, the band around 520 nm (520 nm) includes the 0-2 vibrational satellites of the same modes. Here, we stress that the spectral linewidth in these simulations is fully accounted for by the thermal population of low-frequency soft modes (adding a tiny inhomogeneous contribution to smoothen the spectra). The room temperature spectra show that the $S_0$-$S_1$ excitation is coupled to two molecular vibrations at ≈166 meV (1338.88 cm$^{-1}$) and 205 meV (1653.44 cm$^{-1}$), in excellent agreement with the values inferred from the measured absorption and emission spectra (167 and 210 meV). These are two linear combinations of localized vibrations of the carbon-carbon bonds.

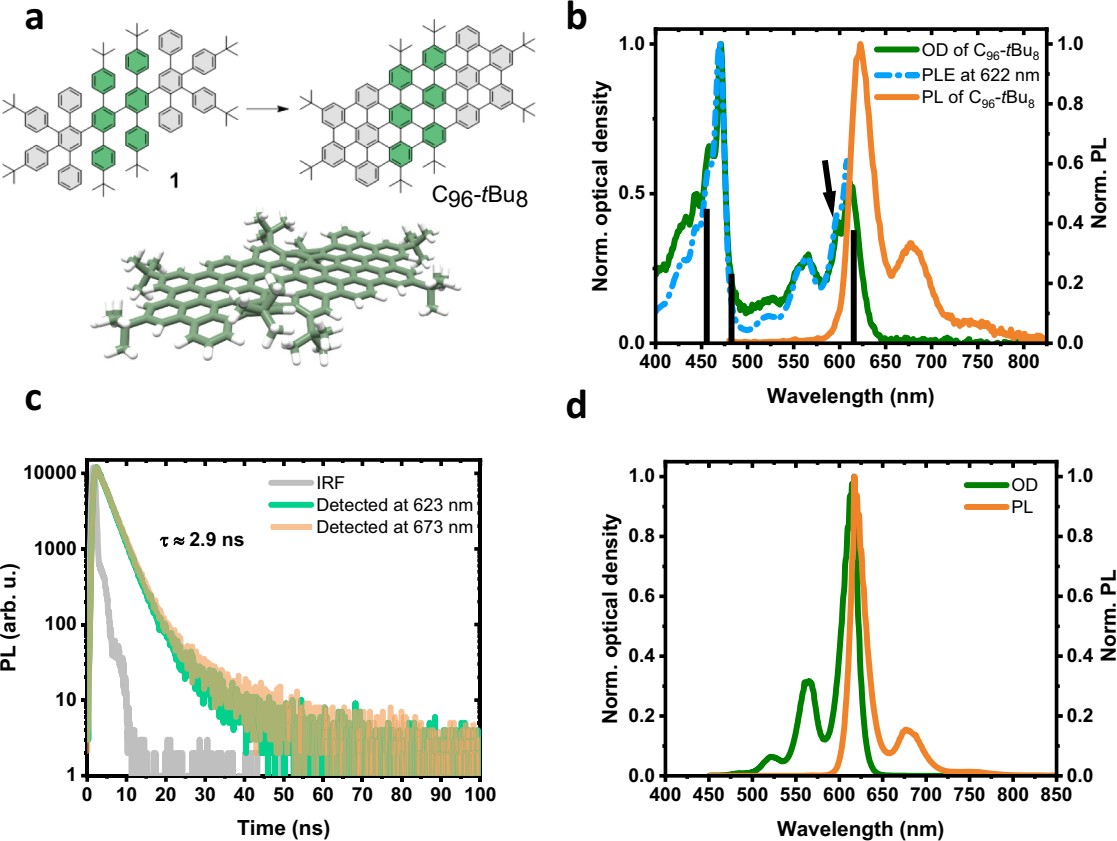

**Fig. 1 | C$_{96}$-$t$Bu$_8$ properties. a** Chemical structure of the polyphenylene dendrimer **1** and the **C$_{96}$-$t$Bu$_8$**. Molecular modeling showing the steric interactions between the *tert*-butyl groups along the long axis of the quantum dots. **b** Absorption (olive), photoluminescence excited at 470 nm (orange) and photoluminescence excitation recorded at 622 nm (dashed blue) spectra of **C$_{96}$-$t$Bu$_8$** in 1,2,4-trichlorobenzene. The black lines correspond to the transitions calculated by DFT/TDDFT with their relative oscillator strength. **c** Time-resolved photoluminescence detected at 623 nm (green) and 673 nm (orange). IRF impulse response function (gray). The lifetime value is extracted from a mono-exponential decay fit. **d** DFT/TDDFT calculations of the first singlet state with vibronic coupling: absorption (olive) and photoluminescence (orange) spectra. Source data are provided as a Source Data file.

The PL spectrum of **C$_{96}$-$t$Bu$_8$** is clean and simple with no signal associated with aggregates and impurities emitting at higher energy[22]. The PL spectrum shows three lines corresponding to the zero-phonon lines and vibronic replicas related to the C=C stretching mode at ≈170 meV. A convincing additional argument suggesting that **C$_{96}$-$t$Bu$_8$** are highly individualized in solution comes from the matching between optical absorption and photoluminescence excitation spectra (PLE) (Fig. 1b). Indeed, the PL signal is related to the emission of monomers while the absorption spectrum is sensitive to all the species in solution. On the contrary, in a first approximation, the PLE spectrum can be understood as the absorption of the emissive species (i.e., monomers in our case). When aggregates are the main component in solution, the absorption spectrum is broadened and thus differs from the PLE spectrum showing sharp lines. This is precisely the behavior of the triangular-shaped **C$_{96}$-(C$_{12}$H$_{25}$)$_6$** GQD[22]. A comparison between the absorption, PL, and PLE of **C$_{96}$-$t$Bu$_8$** and those of **C$_{96}$-(C$_{12}$H$_{25}$)$_6$** is provided in Supplementary Fig. 4. The absorption spectrum of **C$_{96}$-(C$_{12}$H$_{25}$)$_6$** GQD shows broad lines in contrast with the well-defined PLE peaks (Supplementary Fig. 4c). Thus, the perfect match of the PLE and optical absorption spectra for **C$_{96}$-$t$Bu$_8$** means that the absorbing and emitting objects are identical, excluding the substantial existence of aggregates. The time-resolved photoluminescence curves recorded at 623 and at 673 nm for **C$_{96}$-$t$Bu$_8$** show a mono-exponential decay with a lifetime of 2.9 ns (Fig. 1c), suggesting that only one species emits light. All these arguments demonstrate that the solution of **C$_{96}$-$t$Bu$_8$** is composed almost exclusively of monomers. Finally, we were able to measure the absolute

photoluminescence quantum yield (PLQY) of the **C$_{96}$-$t$Bu$_8$** GQD. It reaches an exceptionally high value of 94% (Supplementary Fig. 13), thus highlighting the great potential of GQDs as emitters for optoelectronic applications.

The origin of the line at 597 nm (marked with an arrow in Fig. 1b) in the experimental absorption and PLE spectra is less straightforward. The energy shift from the 0-0 line at 611 nm is of the order of 40 meV. We did not find evidence in our calculations for a vibration mode at such an energy coupled to the S$_0$-S$_1$ excitation, excluding that these lines are vibronic replicas. First, one could hypothesize that this line would arise from the absorption of small aggregates such as dimers or trimers. Nevertheless, no experimental evidence of aggregates in the solution was detected. For instance, experiments as a function of the concentration show the same bands with the same amplitude ratio (see Supplementary Fig. 5b). Finally, we believe that this satellite peak corresponds to the presence of stable conformers. Figure 1a shows the structure of **C$_{96}$-$t$Bu$_8$** with the eight *tert*-butyl groups on the periphery of the GQD. Due to the symmetry of the molecule, six possible conformers (U=up, $t$Bu above the molecular plane; D=down, $t$Bu below the molecular plane) can be considered: UD-UD, UD-DU, UU-UD, UU-DU, UU-UU, and UU-DD (Supplementary Fig. 7). We can safely assume that there is no interconversion among the different conformers, due to their high energy difference (see Supplementary Tables 3–4). The different relative positions of the $t$Bu side groups lead to different degrees of bending of the molecular plane. The distribution in conformations in turn leads to a distribution in gas-phase excitation energies that span a ≈20 meV range, comparable albeit smaller than

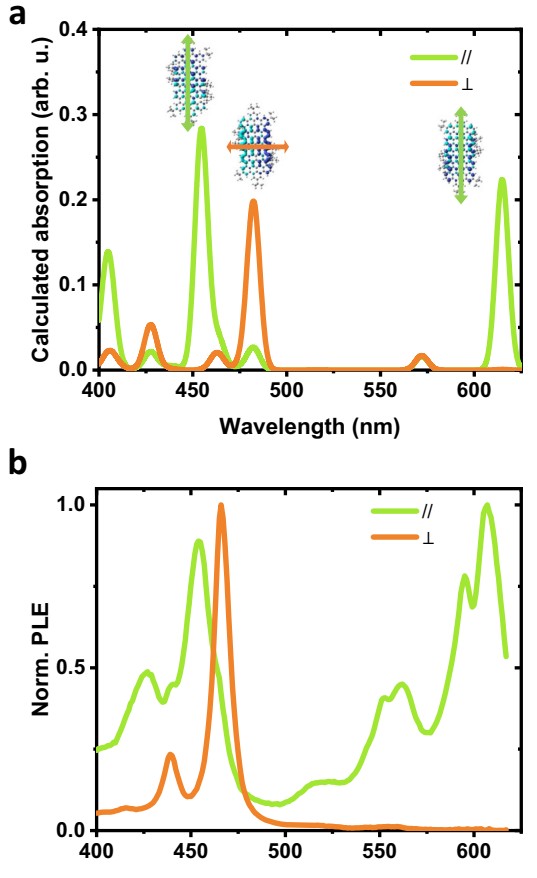

**Fig. 2 | Polarization dependence of the photophysics of C₉₆-*t*Bu₈. a** DFT/TDDFT calculated transitions for polarization along (green) and perpendicular (orange) to the main axis. Transition density distribution and direction of polarization are plotted for illustration. **b** Polarization resolved PLE experiments extracted from anisotropy measurements (see "Methods" for details). The curves correspond to an absorption polarized parallel (green) and perpendicular (orange) to the emission dipole. Source data are provided as a Source Data file.

the measured ≈40 meV difference. We anticipate that the relative stability of these conformers and their optical properties will be sensitive to their liquid or solid environment, which warrants further investigations (see Supplementary Figs. 8, 11, 12).

To get deeper into the photophysics of the GQDs, polarization-resolved experiments have been performed (Fig. 2). The details of the experimental procedure are described in the section "Methods". First, PL experiments resolved in polarization demonstrate that the emission dipole is parallel to the main axis of the C₉₆-*t*Bu₈ GQD, as predicted by theory. The excitation experiments resolved in polarization (Fig. 2b) allow us to observe alternating absorption dipole orientation parallel or perpendicular to the emission dipole. The experimental data perfectly match the predictions by TDDFT, demonstrating the accuracy with which we are able to describe the photophysics of GQDs thanks to their high degree of individualization. Figure 2a and Supplementary Fig. 9 display the TDDFT transition densities associated with the main optical absorption bands. One can see that the first transition at ≈615 nm involves a rearrangement of the density polarized along the main molecular axis of the GQD. In comparison, the second at ≈482 nm is a short-axis polarized excitation. It is followed at ≈455 nm by a third optical excitation that again has its transition dipole moment along the main molecular axis. However, it now involves a local redistribution of the electronic density around the center of the molecule. The excitations remain fully delocalized demonstrating that the *tert*-butyl group

ensures a good individualization without affecting much the planarity and the $sp^2$ system of the GQDs.

Finally, experiments at the single molecule level have been performed on the C₉₆-*t*Bu₈ GQD. Figure 3a shows a photoluminescence raster scan of C₉₆-*t*Bu₈ embedded in a polystyrene film. One can observe localized spots with a size corresponding to the diffraction limit with intensities of a few tens of kcounts s⁻¹. A typical spectrum of a single C₉₆-*t*Bu₈ GQD is displayed in Fig. 3b. It is composed of three lines and is very similar to the one measured in suspension. To push the comparison further, the spectrum of C₉₆-*t*Bu₈ in solution in 1,2,4-trichlorobenzene has been superimposed to the one of the single GQD. The curves fit quasi-perfectly except for the blue side of the single molecule spectrum which is cut by the filter removing the laser excitation. This again demonstrates the homogeneity of the solution arising from both the high degree of individualization and the high purity of C₉₆-*t*Bu₈.

## Tuning the properties by changing the structure

Three other structures were synthesized to demonstrate the control of the excited-state properties of GQDs. The choice was made to keep the $C_{2h}$ symmetry and to change the GQDs' length. Figure 4a shows the C₇₈-*t*Bu₆, C₁₁₄-*t*Bu₁₀, and C₁₃₂-*t*Bu₁₂ GQDs with their respective dendrimer precursors **2, 3**, and **4**. The synthetic scheme is given in Fig. 6 (see "Methods") and it is fully described in the Supplementary Discussion.

Once again, the presence of the *tert*-butyl groups ensures a good solubility and permits the purification of the GQDs and their characterization by MALDI-ToF mass spectrometry, ¹H-¹H and ¹H-¹³C 2D NMR experiments NMR and cyclic voltammetry (see Supplementary Information). After purification, the MALDI-ToF spectrum of C₇₈-*t*Bu₆ showed two sets of signals corresponding to the monomer and the dimer with a larger intensity for the latter, suggesting the possibility of the C₇₈-*t*Bu₆ GQD to form small aggregates (Supplementary Figs. 2a, 5 and 11). The dimer signals are not observed for the C₉₆-*t*Bu₈, C₁₁₄-*t*Bu₁₀, and C₁₃₂-*t*Bu₁₂ GQDs (Supplementary Fig. 2b–d). Despite their bigger sizes C₉₆-*t*Bu₈, C₁₁₄-*t*Bu₁₀, and C₁₃₂-*t*Bu₁₂ GQDs do not form multimeric species in solution at spectroscopic concentration. In all cases, the main peak in the MALDI-ToF spectra corresponds to the desired product where the isotopic distribution is in accordance with the simulated one.

Figure 4b displays the absorption spectra of the three GQDs in 1,2,4-trichlorobenzene. The PL and time-resolved PL spectra of C₇₈-*t*Bu₆, C₁₁₄-*t*Bu₁₀ and C₁₃₂-*t*Bu₁₂ GQDs are given in Supplementary Fig. 11. First, one can observe well-defined and sharp lines indicating a high degree of solubility whatever the size of the GQD. Moreover, the 0-0 line shifts toward the red while increasing the length of the GQD, starting from 554 nm for the C₇₈-*t*Bu₆ GQD, 611 nm for the C₉₆-*t*Bu₈, 652 nm for the C₁₁₄-*t*Bu₁₀ and 688 nm for the C₁₃₂-*t*Bu₁₂. The possibility of tuning the optical transitions by more than 100 nm is a great asset for using GQD in optoelectronic devices. This is again a strong indication that the excited states of the GQDs remain fully delocalized. While emitting in the red, the C₁₁₄-*t*Bu₁₀ and C₁₃₂-*t*Bu₁₂ GQDs show very high PLQY (91% and 88%, respectively). Figure 4c shows the energy variation of the different optical transitions with the size of the GQD. Again, there is a very good match between the theoretical predictions and the experiments. Looking at the absorption spectra, one can observe a variation in the weight of the oscillator strengths of the different transitions. It originates from transition densities associated with the different optical transitions in the different GQDs. The higher the number of atoms in the GQDs, the lower the ratio between the oscillator strengths / integrated intensity to the second and first optical transitions (Fig. 4d). As before, this behavior is predicted by our calculations. Indeed, one can see (Supplementary Fig. 9) that the lowest energy transition is highly delocalized and involves a long-range rearrangement of the electron density along the main axis (electron

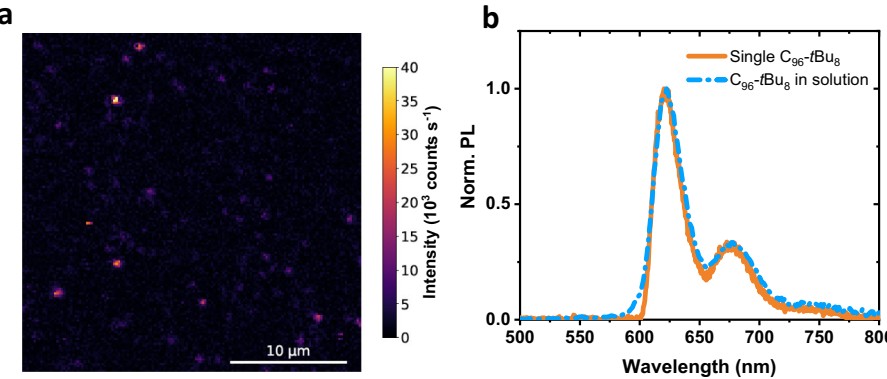

**Fig. 3 | Single molecule experiments on C₉₆-*t*Bu₈. a** Photoluminescence map of **C₉₆-*t*Bu₈** in polystyrene. **b** Normalized photoluminescence spectrum of a single **C₉₆-*t*Bu₈** (orange) excited at 594 nm. The dashed blue curve corresponds to a spectrum of **C₉₆-*t*Bu₈** in 1,2,4-trichlorobenzene. Source data are provided as a Source Data file.

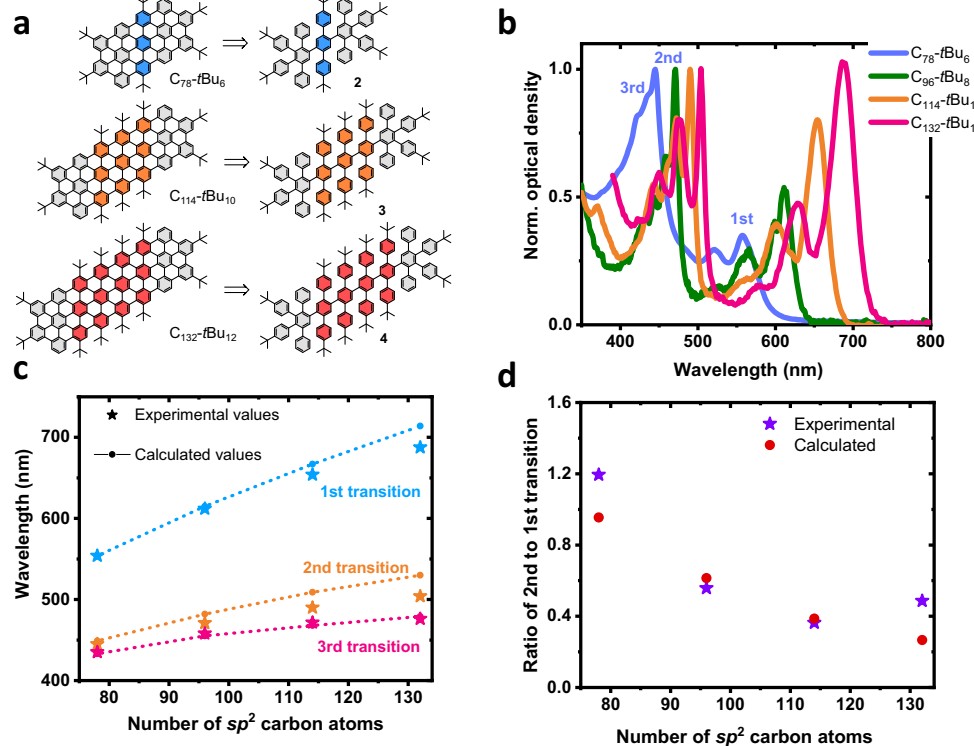

**Fig. 4 | Structure of C₇₈-*t*Bu₆, C₁₁₄-*t*Bu₁₀ and C₁₃₂-*t*Bu₁₂ GQDs, absorption spectra of the GQDs, and comparison with theory. a** Dendrimers and their corresponding GQDs with 78, 114, and 132 $sp^2$ carbon atoms. **b** Absorption spectra of the four graphene quantum dots (**C₇₈-*t*Bu₆** – blue, **C₉₆-*t*Bu₈** – green, **C₁₁₄-*t*Bu₁₀** – orange and **C₁₃₂-*t*Bu₁₂** – pink). **c** Variation of the transition energies, marked as (1st, 2nd and 3rd) in the absorption spectra, as a function of the number of atoms; the stars correspond to the experimental values and the lines with dots correspond to calculated values (blue and dotted blue: first transition, orange and dotted orange: second transition and pink and dotted pink: third transition). **d** Variation of the ratio second to first transition intensity as a function of the number of atoms (purple stars: experimental values, red dots: calculated values). Source data are provided as a Source Data file.

densities on one side, hole on the other). On the contrary, the second transition involves rearranging the electron density over a shorter distance and in the perpendicular direction. This explains the relative change in intensity with the increase in the longitudinal length of the GQDs.

## GQDs aggregation

To understand the different aggregation behaviors among the GQDs, we performed calculations of the Potential of Mean Force (PMF) curves corresponding to **C₇₈-*t*Bu₆** and **C₉₆-*t*Bu₈** dimer formation, using Umbrella Sampling (US) and the Weighted Histogram Analysis Method (WHAM)[48] (Fig. 5). Model systems contain two GQDs in their most stable conformation (2 **C₇₈-*t*Bu₆** U-U and U-D or 2 **C₉₆-*t*Bu₈** UD-DU and UD-UD, see Supplementary Figs. 7 and 15 and Supplementary Tables 5 and 6 for details on the conformers) solvated in a cubic box of 1,2,4-trichlorobenzene molecules. Systems were first equilibrated for 80 ns in the NPT ensemble at 293.15 K and 1 atm, and a harmonic restraint between the two GQDs Center Of Mass (COM) was progressively switched on to keep them 20 Å apart from each other. Then, they were pushed towards each other down to 2 Å, and pulled apart back to 20 Å. Two pulling curves were computed, starting from local minima 2 and 3 reached during the pushing, denoted as min.2 and min.3 in Fig. 5

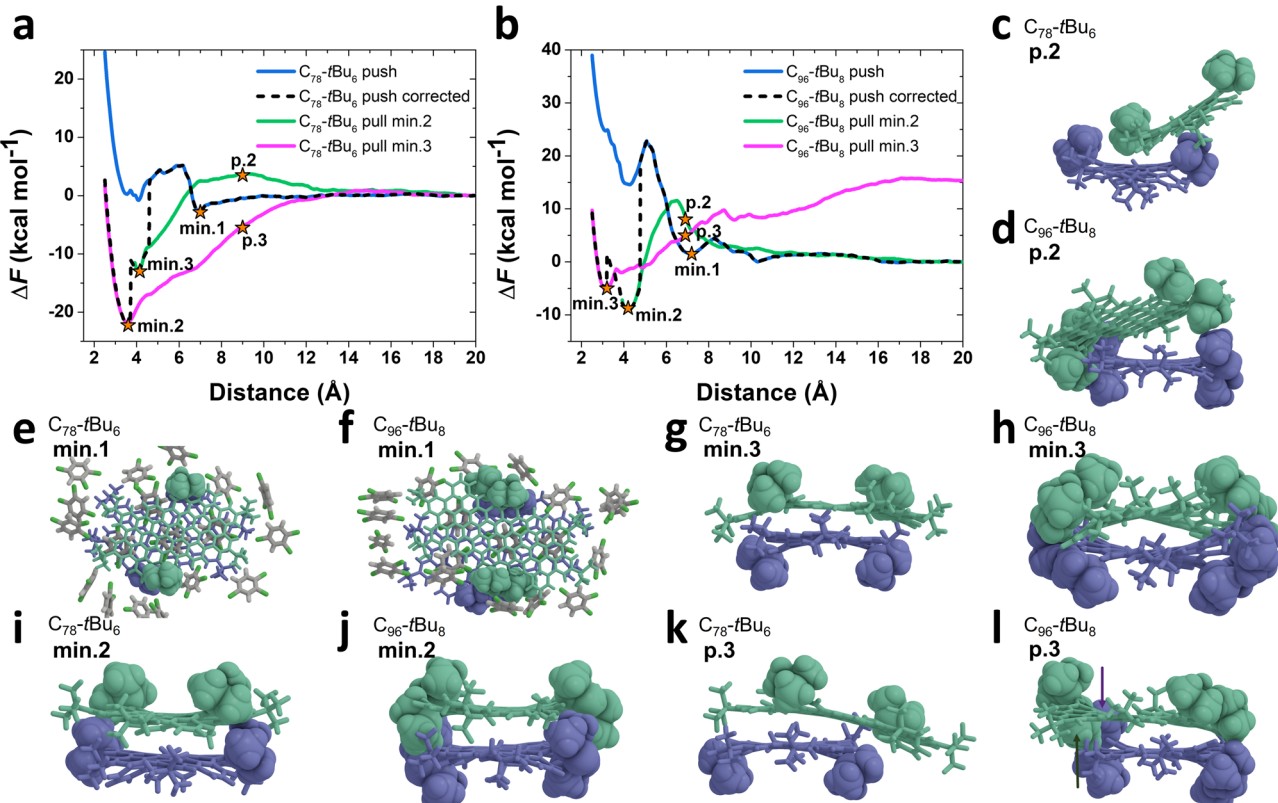

**Fig. 5 | Simulation of aggregates for C$_{78}$-$t$Bu$_6$ and C$_{96}$-$t$Bu$_8$.** Potential of Mean Force curves for **C$_{78}$-$t$Bu$_6$** (**a**) and **C$_{96}$-$t$Bu$_8$** (**b**). The plots show the variation of free energy as a function of the distance between the COM of the two GQDs. Minima in the curves are labeled as min.1, min.2, and min.3, and specific points of interest along the trajectory are labeled as p.2 and p.3. **c**–**l** Snapshots of the **C$_{78}$-$t$Bu$_6$** and **C$_{96}$-$t$Bu$_8$** dimer conformation at the minima and specific points of interest. Except for conformations at min.1, solvent molecules were omitted for clarity (they are represented as sticks otherwise, with C in gray, H in white and Cl in green). Molecules forming the dimer are colored, respectively in green and blue to improve visibility. Atoms in *tert*-butyl groups are represented as Van der Waals spheres while the rest of the GQDs are represented with sticks.

(visualizations were made with the Visual Interactive Analysis of Molecular Dynamics VIAMD software[49]).

The push curves for **C$_{78}$-$t$Bu$_6$** and **C$_{96}$-$t$Bu$_8$**, suffer from artifacts linked to finite sampling during the free energy calculation. They show a first local minimum at around 7 Å, followed by a large potential barrier to reach a second minimum at around 4 Å: the first minimum corresponds to GQDs in a cofacial stack separated by one layer of solvent (Fig. 5, min.1). The solvent needs to be expelled to reach the second minimum where the GQDs are in direct contact (Fig. 5, min.2). This path is not the most favorable to form the dimer (as obtained from the pull curves, *vide infra*), and more importantly, the sudden solvent removal leads to a conformational dissipation that is not captured in the PMF curves. This can be evidenced by comparing the US samplings as a function of the reaction coordinate, i.e., the GQDs COM distance (Supplementary Fig. 16) and other hidden variables such as the GQDs core COM distance (Supplementary Fig. 17). While the samplings with the reaction coordinate appear continuous, the samplings with the core COM distance show some discontinuities for **C$_{78}$-$t$Bu$_6$** push and **C$_{96}$-$t$Bu$_8$** push, indicating some brutal conformational dissipations. The "corrected" push curves show the location of these dissipations along the free energy curves. For **C$_{78}$-$t$Bu$_6$**, they were constructed by setting the reference zero free energy in the unbound states (two U-U **C$_{78}$-$t$Bu$_6$** apart from each other) and by matching the minima explored during push, pull from minimum 2 and pull from minimum 3. The same method was used for **C$_{96}$-$t$Bu$_8$**, except that the unbound reference energy for the pull curve from minimum 3 is shifted upward because the GQD conformations, in this case, are UU-DU and UU-UD, see below (the shift

has been calculated by the classical force-field in the solvated system, and is rather similar to the DFT shift obtained on isolated molecules: $E_{UU-DU} + E_{UU-UD} - 2 \times E_{UD-DU} \approx 16$ kcal mol$^{-1}$). These finite sampling effects are well known, and would require increasing the sampling time and/or resorting to more sophisticated PMF calculations to obtain accurate quantitative results[50]. Yet, to get a qualitative picture of the GQDs aggregation we will focus on the pull curves in the following.

For **C$_{78}$-$t$Bu$_6$**, the minimum 2 corresponds to a dimer where the *tert*-butyl groups of one GQD point outward the dimer, and the *tert*-butyl groups of the other one point inward (Fig. 5). These inward groups stabilize the COM distance around 4 Å. During the dimer dissociation (or formation) the top **C$_{78}$-$t$Bu$_6$** has to "climb" over the *tert*-butyl pointing inward, (Fig. 5, p.2). As a result, a small potential barrier for the aggregation, which might be crossed by thermal agitation, can be seen on the **C$_{78}$-$t$Bu$_6$** PMF pull min.2. A more stable dimer, corresponding to minimum 3, can be reached if all the *tert*-butyl groups now point outward (Fig. 5, min.3). In this conformation, the two **C$_{78}$-$t$Bu$_6$** can gently slide on top of each other to dissociate. Most importantly, min.3 is a deep minimum with close intermolecular separation (<4 Å) and stabilization energy > 20 kcal mol$^{-1}$ compared to non-interacting molecules.

**C$_{96}$-$t$Bu$_8$** shows the same behavior with more pronounced effects. Minimum 2 is also around 4 Å, due to half of the *tert*-butyl groups pointing inward. During the dimer dissociation or formation, one of the *tert*-butyl of the top molecule has now to "climb" over a *tert*-butyl of the bottom molecule (at p.2 on Fig. 5). As a result, there is a high potential barrier that prevents GQDs aggregation. Another minimum

corresponding to a shorter COM distance, min.3, can be reached if all *tert*-butyl groups point outward. The free energy value of this minimum is probably not very accurate, as it can be seen from the small discontinuity in Supplementary Fig. 17—$C_{96}$-*t*$Bu_8$ pull min.3. Yet, it is likely higher than minimum 2 due to the very unstable $C_{96}$-*t*$Bu_8$ conformations (UU-UU for both molecules) adopted by the GQDs. Indeed, during the dimer dissociation, the conformations change toward UU-DU for the top molecule and UU-UD for the bottom molecule, as highlighted by the arrows on the snapshot corresponding to point p.3 for $C_{96}$-*t*$Bu_8$. Interestingly, since on the right side of the dimer all *tert*-butyl groups point outward, the molecule can still slide on top of each other, and the $C_{96}$-*t*$Bu_8$ pull min.3 curve does not show any potential barrier that might impede the GQDs aggregation. However, these conformers do not exist in solution, as highlighted by the high free energy value in the unbound state. Overall, the picture that emerges from the simulations is the following. As long as the molecules can adopt a conformation with all (most) *tert*-butyl groups pointing outward, aggregation may proceed through the sliding of one molecule on top of the other. This situation can happen for any couple of $C_{78}$-*t*$Bu_6$ conformers: considering any combination of U-U and U-D, there is always a chance to have a dimer side with all *tert*-butyl pointing outward. On the contrary, this is not possible in the two most stable conformers of $C_{96}$-*t*$Bu_8$, UD-DU and UD-UD, since a *tert*-butyl pointing outward is always sitting next to a *tert*-butyl pointing inward on the same side of the molecule. Such an aggregation pathway would, in principle, be possible for the other conformers (with adjacent U or D side groups). However, these are much less stable as they involve much larger intramolecular strain and are very likely not present in solution.

In conclusion, aggregation can be avoided in GQDs by adding *tert*-butyl on the molecules if the most stable conformers show adjacent *tert*-butyl groups pointing in opposite directions, as turns out to be the case in $C_{96}$-*t*$Bu_8$ but not in $C_{78}$-*t*$Bu_6$.

## Discussion

Here, we described a family of graphene quantum dots bearing bulky solubilizing groups that permit to improve significantly their individualization in solution without affecting the delocalization of the π-conjugated systems. We explored the GQDs' photophysics and gave an accurate description of their properties by combining optical experiments down to the single molecule level and TDDFT calculations. In particular, we show that calculations and experiments agree on the prediction of the variation of the excited state energy and of their oscillator strength with the size of the GQD. Calculations also describe well the measured polarization dependence of the lowest excited states. In the future, the accuracy between theory and experiments could be further investigated, for instance, using low temperature spectroscopy revealing electron-vibration coupling and its evolution as a function of the GQD structure. Finally, by introducing quantum emitters with well-understood properties, tunable by chemistry, and readily dispersible and individualized, our work opens the way for using GQDs as emitters for several applications, especially in quantum technologies. First, integrated in photonics networks or for quantum key distribution where bright quantum emitters emitting single photon on-demand are highly desirable. Then, taking advantage of their one atom layer thickness and thus their huge sensitivity to the local environment, GQDs are also very promising for quantum sensing applications.

## Methods
### Techniques
NMR spectra were recorded with a Bruker Avance Neo 400 (400 MHz) instrument or on a Bruker Avance II 600 (600 MHz) instrument, with solvent used as an internal reference. MALDI-TOF mass spectra were recorded on a Bruker Autoflex speed or on a Bruker UltrafleXtreme; the samples were analyzed in DCTB (*trans*-2-[3-(4-*tert*-butylphenyl)-2-methyl-2-propenylidene]malononitrile) matrix. Absorption spectra were recorded in quartz cuvettes on a Perkin Elmer Lambda 900 UV-Vis-NIR spectrophotometer. Thin layer chromatography (TLC) was performed on silica gel 60 F254 (Merck) precoated aluminum sheets. Column chromatography was performed on Merck silica gel 60 (0.063−0.200 mm) or puriFlash® Si-HP 60 Å 15 or 30 μm columns with an Interchim puriFlash 430. SEC was performed on Bio-Beads S-X1 or Bio-Beads S-X3 with THF as eluent.

### Materials
Pd(OAc)$_2$ (Sigma, 97%), SPhos (TCI Europe, >98%), K$_3$PO$_4$ (Sigma, ≥98%), ICl (Sigma, 1 M solution in CH$_2$Cl$_2$), Pd(PPh$_3$)$_2$Cl$_2$, (Sigma, 98%), CuI (Sigma, 98%), ethynyltrimethylsilane, (Sigma, 98%), tetra-butylammonium fluoride (Sigma, 1 M THF), *o*-xylene (Sigma, anhydrous 97%), FeCl$_3$, (Sigma, anhydrous for synthesis), CH$_3$NO$_2$ (Sigma, absolute, over molecular sieve, ≥98.5%), B$_2$Pin$_2$, (Combi-Blocks, 98%), Pd(dppf)Cl$_2$ (Sigma, not specified), KOAc (Sigma, anhydrous ≥99%), 1,4-dioxane (Thermo Scientific Chemicals, anhydrous 99.8%), Pd(PPh$_3$)$_4$ (Sigma, 99%), 2,3-dichloro-5,6-dicyano-1,4-benzoquinone (Thermo Scientific Chemicals, ≥98%), triflic acid (Sigma, ≥ 99%), AgNO$_3$ (Sigma, ≥99%), and tetra-n-butylammonium hexafluorophosphate (Sigma, 98%) were used as received. 1,4-diiodo-2,5-bis(-trimethylsilyl)benzene[51] and 2,5-diphenyl-3,4-di(4-*tert*-butylphenyl)-cyclopentadien-1-one[52] were synthesized according literature procedures. Solvents were purchased from SDS Carlo Erba, Aldrich, and Fisher Scientific and were used as received. For synthesis, unstabilized CH$_2$Cl$_2$ (CaH$_2$, N$_2$), toluene (K/benzophenone, N$_2$), THF (K/benzophenone, N$_2$), Et$_3$N (CaH$_2$, N$_2$), and Et$_2$O (CaH$_2$, N$_2$) were distilled before use.

### Synthesis
The synthesis of the polyphenylene dendrimers **1**, **2**, **3** and **4** precursors of the GQDs is depicted in Fig. 6. It starts with the Suzuki-Miyaura coupling between 1,4-diiodo-2,5-bis(trimethylsilyl)benzene (**5**)[51] and 4-*tert*-butylphenylboronic acid in the presence of Pd(OAc)$_2$ and SPhos (2-dicyclohexylphosphino-2′,6′-dimethoxybiphenyl) to give the bis(trimethylsilyl)-terphenyl derivative **6**. This molecule serves as common base for the synthesis of the three nanoparticles. Terphenyl **6** is converted into diiodo-terphenyl derivative **7** by reaction with iodine monochloride (ICl). The Sonogashira cross-coupling between **7** and ethynyltrimethylsilane in the presence of Pd(PPh$_3$)$_3$Cl$_2$ and CuI gives **8**, which can be deprotected with 1 M THF solution of tetra-n-butylammonium fluoride (TBAF) to yield the alkyne terphenyl core. The Diels-Alder reaction between the alkyne terphenyl core and the 2,5-diphenyl-3,4-di(4-*tert*-butylphenyl)-cyclopentadien-1-one (**9**)[52] gives **2**, the precursor of the $C_{78}$-*t*$Bu_6$.

For the synthesis of the core of the $C_{96}$-*t*$Bu_8$ GQD, the bis(-trimethylsilyl)-terphenyl derivative **6** is mono iodinated with one equivalent ICl in the presence of AgBF$_4$ to give **10**, which is borylated with bis(pinacolato)diboron in the presence of [1,1′-bis(diphenylphosphino)-ferrocene] dichloropalladium (II) (Pd(dppf)Cl$_2$) under Miyaura conditions to give **11**. The Suzuki-Miyaura coupling between **10** and **11** in the presence of tris(dibenzylideneacetone)-dipalladium (Pd$_2$(dba)$_3$) and SPhos leads to the diterphenyl derivative **12**, which is converted into **13** by reaction with ICl. The Sonogashira cross-coupling between **13** and ethynyltrimethylsilane gives **14**, which can be deprotected with TBAF to yield the alkyne diterphenyl core. The Diels-Alder reaction between the alkyne diterphenyl core and the cyclopentadienone **9** gives **1**, the precursor of the $C_{96}$-*t*$Bu_8$ GQD.

The core of the $C_{114}$-*t*$Bu_{10}$ GQD is synthesized following the same methodology; the Suzuki-Miyaura coupling between **7** and **11** leads to the triterphenyl derivative **15**, which is converted into **16** by reaction with ICl. The Sonogashira cross-coupling between **16** and ethynyltrimethylsilane gives **17**, which can be deprotected with TBAF to yield the alkyne triterphenyl core. The Diels-Alder reaction between the

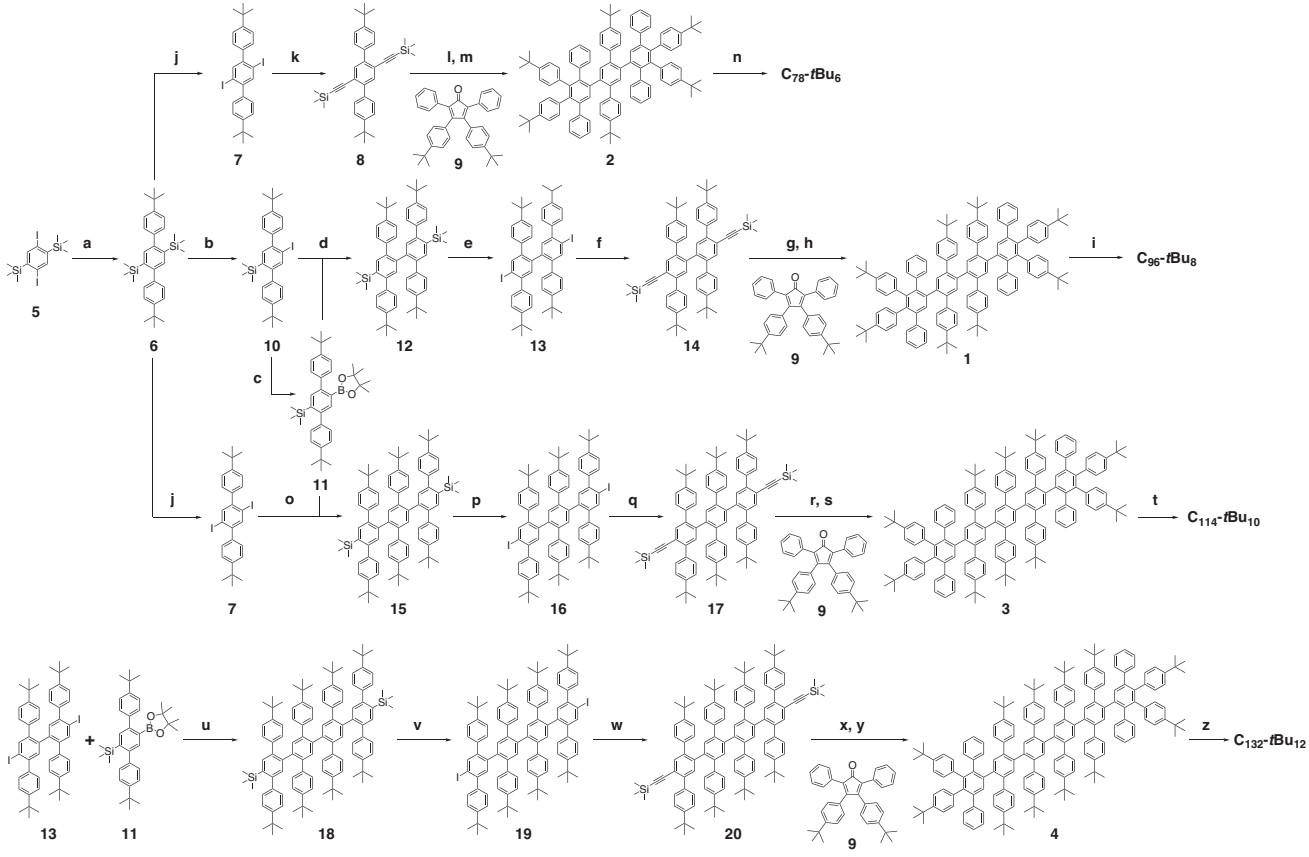

**Fig. 6 | Synthesis of intermediates and GQDs. (a)** Pd(OAc)$_2$, SPhos, K$_3$PO$_4$, toluene, water, 90 °C, 12 h, 82%; **(j)** 1 M ICl solution in CH$_2$Cl$_2$, CH$_2$Cl$_2$, rt, overnight, 100%; **(k)** Pd(PPh$_3$)$_2$Cl$_2$, CuI, ethynyltrimethylsilane, toluene, Et$_3$N, 80 °C, 16 h, 38%; **(l)** TBAF, THF, rt, 2 h; **(m)** o-xylene, 180 °C, 16 h, 99%; **(n)** FeCl$_3$, CH$_2$Cl$_2$/CH$_3$NO$_2$, rt, 25%; **(b)** AgBF$_4$, 1 M ICl solution in CH$_2$Cl$_2$, MeOH/THF, 0 °C, rt, 1 h 45 min, 62%; **(c)** B$_2$Pin$_2$, Pd(dppf)Cl$_2$, KOAc, 1,4-dioxane, 95 °C, 36 h, 58%; **(d)** Pd(OAc)$_2$, SPhos, K$_3$PO$_4$, toluene, water, 90 °C, 36 h 83%; **(e)** 1 M ICl solution in CH$_2$Cl$_2$, CH$_2$Cl$_2$, rt, overnight, 88%; **(f)** Pd(PPh$_3$)$_2$Cl$_2$, CuI, ethynyltrimethylsilane, toluene, Et$_3$N, 80 °C, overnight, 87%; **(g)** TBAF, THF, rt, 1 h; **(h)** o-xylene, 175 °C, 16 h, 43%; **(i)** FeCl$_3$, CH$_2$Cl$_2$/CH$_3$NO$_2$, rt, 8% (after extensive purification); **(o)** Pd(PPh$_3$)$_4$, K$_2$CO$_3$, toluene, ethanol, water, 80 °C, 21 h, 65%; **(p)** 0.5 M ICl solution in CH$_2$Cl$_2$, CH$_2$Cl$_2$, rt, overnight, 85%; **(q)** Pd(PPh$_3$)$_2$Cl$_2$, CuI, ethynyltrimethylsilane, toluene, Et$_3$N, 85 °C, 21 h, 35%; **(r)** TBAF, THF, rt, 2 h; **(s)** o-xylene, 180 °C, 16 h, 72%; **(t)** FeCl$_3$, CH$_2$Cl$_2$/CH$_3$NO$_2$, rt, 6% (after extensive purification). **(u)** Pd(PPh$_3$)$_4$, K$_2$CO$_3$, toluene, ethanol, water, 85 °C, 20 h, 72%; **(v)** 1 M ICl solution in CH$_2$Cl$_2$, CH$_2$Cl$_2$, rt, overnight, 93%; **(w)** Pd(PPh$_3$)$_2$Cl$_2$, CuI, ethynyltrimethylsilane, toluene, Et$_3$N, 85 °C, 21 h, 45%; **(x)** TBAF, THF, rt, 2 h; **(y)** o-xylene, 180 °C, 18 h, 59%; **(z)** 2,3-dichloro-5,6-dicyano-1,4-benzoquinone, CH$_2$Cl$_2$, triflic acid, 0 °C, 40 min, 33%.

alkyne triterphenyl core and the cyclopentadienone **9** gives **3**, the precursor of the C$_{114}$-$t$Bu$_{10}$ GQD.

For C$_{132}$-$t$Bu$_{12}$ GQD, the Suzuki-Miyaura coupling between **13** and **11** leads to the tetraterphenyl derivative **18**, which is converted into **19** by reaction with ICl. The Sonogashira cross-coupling between **19** and ethynyltrimethylsilane gives **20**, which can be deprotected with TBAF to yield the alkyne triterphenyl core. The Diels-Alder reaction between the alkyne triterphenyl core and the cyclopentadienone **9** gives **4**, the precursor of the C$_{132}$-$t$Bu$_{12}$ GQD.

The polyphenylene precursors **1, 2** and **3** are finally oxidized in the presence of FeCl$_3$ in a mixture of dichloromethane and nitromethane under argon flux to give the GQDs C$_{96}$-$t$Bu$_8$, C$_{78}$-$t$Bu$_6$, and C$_{114}$-$t$Bu$_{10}$, respectively. The polyphenylene precursor **4** is converted into C$_{132}$-$t$Bu$_{12}$ by oxidation with DDQ (2,3-dichloro-5,6-dicyano-1,4-benzoquinone) in the presence of triflic acid. Compared to the previously described C$_{96}$(C$_{12}$H$_{25}$)$_6$ GQD[21,46], the oxidation of the *tert*-butyl-containing precursors into GQD happened readily fast. We attributed this enhanced reactivity to the higher electron donor character of the *tert*-butyl groups compared to alkyl chains. It is worth mentioning that the complete oxidation of **1, 3**, and **4** into C$_{96}$-$t$Bu$_8$, C$_{114}$-$t$Bu$_{10}$ and C$_{132}$-$t$Bu$_{12}$, respectively, was successful despite the presence of two, three or four successive bulky *tert*-butyl groups along the long axis of the nanoparticles.

## Electrochemical experimental details

For all the cyclic voltametric (CV) measurements, analytical grade tetra-n-butylammonium hexafluorophosphate (TBAH) was used as received as a supporting electrolyte; anhydrous grade 1,2-dichlorobenzene (Sigma-Aldrich, 99%) was used as solvent as received. Electrochemical experiments were carried out in a three electrodes single-compartment cell using platinum as working and counter electrodes and a silver wire in 10 mM AgNO$_3$ in acetonitrile as reference electrode under N$_2$ and at a scan rate of 100 mV s$^{-1}$. The $E_{1/2}$ potentials have been directly obtained from cyclic voltametric curves as averages of the cathodic and anodic peak potentials; the ferrocinium/ferrocene couple (Fc$^+$/Fc) was used as reference potential (+0.42 V vs. SCE at 298 K). The CV were recorded with a VSP bipotentiostat (Bio-Logic SAS).

## Photoluminescence experiments in solution

Measurements in solution are performed using a commercial spectrometer (FluoroMax+, Horiba). Quartz cuvettes (Hellma Analytics, Quartz Glass High performance cuvette) are filled with sample solution and placed in the excitation beam path. A Xenon lamp, coupled with a monochromator, is used for excitation in PL and PLE measurements. Time-resolved PL measurements use a solid-state pulsed diode (NanoLED) with excitation at 483 nm (pulse width below 100 ps, 100 MHz repetition rate). Signal detection is performed using a

photon-counting detector (R928P photomultiplier tube), coupled with a monochromator for wavelength selection.

## Experiments in polarization

Polarizers are added to the excitation and emission beam paths for polarization-resolved experiments in solution. The anisotropy factor $r$ is computed from the measurement of the PLE spectrum collected at the same emission wavelength for four different polarizer orientations (each polarizer is oriented either horizontally H or vertically V):

$$r = \frac{I_{VV} - GI_{VH}}{I_{VV} - 2GI_{VH}}, \text{ with } G = \frac{I_{HV}}{I_{HH}} \tag{1}$$

where $I_{AB}$ corresponds to the PLE spectrum with the excitation polarizer in orientation $A$ and the emission polarizer in orientation $B$.

The anisotropy factor is decomposed into parallel and perpendicular contributions using:

$$r = f_\parallel r_\parallel + f_\perp r_\perp \tag{2}$$

where $r_\parallel$ and $r_\perp$ the anisotropy value for respectively perfectly parallel or perpendicular transitions relatively to the emission dipole, and $f_\parallel + f_\perp = 1$. The value of $r_\parallel$ is evaluated to $r_{max}$ the maximum value reached by $r$, to consider possible depolarization effects, and $r_\perp = -\frac{r_{max}}{2}$. The parallel $I_\parallel$ and perpendicular $I_\perp$ components of the PLE spectrum are computed considering:

$$I_\parallel = f_\parallel I_{total} = \frac{r - r_\perp}{r_\parallel - r_\perp} I_{total} \tag{3}$$

$$I_\perp = f_\perp I_{total} = \frac{r_\parallel - r}{r_\parallel - r_\perp} I_{total} \tag{4}$$

with $I_{total}$ the non-polarized PLE spectrum.

## Determination of the photoluminescence quantum yield

PLQY measurement are carried on a FS5 spectrofluorometer from Edinburgh Instruments, using the SC-30 integrating sphere. The manufacturer provides quartz cuvettes dedicated to the integrating sphere. A monochromator selects the excitation wavelength from a Xenon lamp. Light emitted and diffused by the sample is then collected by the integrating sphere and sent to a second monochromator, allowing to select the collection wavelength, before being sent to a photomultiplier tube. Two steps are involved in the measurement of quantum yield. First, a cuvette filled with solvent is introduced in the sphere. The excitation monochromator is set to illuminate the sample at the desired excitation wavelength (Supplementary Fig. 13). The whole excitation and emission range is then scanned by the collection monochromator (typically from 400 to 900 nm), with a bandwidth ratio of ten between excitation and collection. Typically, excitation bandwidth is 2 nm and collection bandwidth is 0.2 nm. This first step is repeated with a cuvette filled with a solution containing the molecule. For optimal measurement of the PLQY, optical density of the sample must be between 0.1 and 0.3. These two steps yield two spectra, denoted R and S for the reference and the sample respectively. We denote $Ex_R$ and $Ex_S$ the integrated spectra around the excitation wavelength, and $Em_R$ and $Em_S$ the integrated spectra on the emission range. The PLQY ($\Phi$) is then calculated as follow:

$$\Phi = (Em_S - Em_R)/(Ex_R - Ex_S) \tag{5}$$

To test the accuracy of the protocol, we measure the PLQY of fluoresceine, whose PLQY is known to be $0.925 \pm 0.015$[53]. We measured a quantum yield of 0.9118 on a fluoresceine sample in a 0.1 M NaOH solution with an optical density of 0.24, providing confidence in the good quality of the measurements from our setup.

## Samples for single molecule experiments

Samples for single-object experiments are prepared from a mixture of $C_{96}$-$t$Bu$_8$ GQD solution and polystyrene in 1,2,4-trichlorobenzene solution (yielding a concentration in polystyrene of 10 wt%). The mixture is spin-coated on glass substrate (Schott Nexterion) previously plasma-cleaned for 5 min. The sample is then dried on a hot plate (90 °C) for 1 h.

## Single-molecule experiments

Single-object experiments use a home-built micro-PL setup under ambient conditions. The sample is excited using a 594 nm CW diode laser (Cobolt Mambo 100), which is focused on the sample using an oil-immersive microscope objective (NA = 1.42, Olympus PLAPON 60X), mounted on XYZ piezoelectric translational stages (Mad City Labs Inc.) for sample scanning. Luminescence signal is collected using the same objective and focused on a 50 μm pinhole for confocal selection. Spectral selection is performed to remove the excitation beam using a dichroic mirror (zt 594 RDC, Chroma) and a long-pass filter (FELH0600, Thorlabs). After confocal selection, the signal is either directed to silicon-based avalanche photodiodes (SPCM-AQR-13, PerkinElmer) for raster scan measurements, or to a monochromator (SP-2350, Princeton Instruments) coupled to a LN-cooled CCD camera (PyLoN:100BR eXcelon, Princeton Instruments) for spectral measurements. Interface of the translational stages and the detectors with computer uses an acquisition card (PCIe-6323, National Instruments). Raster scans and result display are performed using a python-based suite[54].

## Computational details

The different GQDs have been studied at the DFT level of theory with the HSE functional and 6-31 G(d,p) basis set, in gas phase, within the Gaussian16 software. Benchmark calculations with different long-range corrected functionals have been performed, and the results are reported in Supplementary Fig. 8. The geometry optimization has been performed for each structure, followed by TDDFT single point calculations to obtain the absorption spectra. The effect of the solvent on absorption has been investigated and is reported in Supplementary Information. Emission spectra were obtained by optimizing the first excited state. Frequencies in both ground and excited states were computed to obtain the vibronic spectra. Transition densities were obtained by the Nancy-EX post-processing software[55,56].

## Vibronic progression method

To model the vibronic progression in the optical absorption and emission spectra, we start with the calculation of the exciton-phonon coupling constants and related Huang-Rhys factors, $S_j$, applying an undistorted (that is we assume the same normal modes in the ground, $g$, and lowest excited state, $e$, that means that we ignore Duschinsky matrix rotation effects) but displaced (or shifted) harmonic oscillator model.

Thus, we project the displacements between the two optimized geometries in a Cartesian framework onto the ground-state (for absorption) or excited-state (for emission) normal modes:

$$d_j = Q_j(e) - Q_j(g) \tag{6}$$

The dimensionless Huang-Rhys factor for normal mode $j$ is given by

$$S_j = \frac{1}{2\hbar} \omega_j d_j^2 \tag{7}$$

with $\omega_j/2\pi$ being the corresponding normal-mode frequency. We then compute the optical absorption spectrum in the Franck-Condon approximation using[57]:

$$\langle\alpha(\omega)\rangle = \omega \int_{-\infty}^{\infty} dt\, e^{it(\omega_{ge}-\omega)-\frac{D_{ge}t^2}{4}-\gamma_{ge}|t|} \exp\left[-\sum_j S_j\left\{\left(2\bar{n}_j+1\right) - \left(\bar{n}_j+1\right)e^{it\omega_j}-\bar{n}_j e^{-it\omega_j}\right\}\right] \quad (8)$$

where $\omega_{ge}$ represents the adiabatic (0-0) transition energy from the ground state to the excited state, $\bar{n}_j=(e^{\frac{\hbar\omega_j}{kT}}-1)^{-1}$ is the Bose-Einstein population of the vibrational normal modes at temperature $T$, and $\gamma_{ge}(D_{ge})$ quantifies (in)homogeneous broadening. We set $\gamma_{ge}$ to zero and $D_{ge}$ to a very small ($10^{-4}\,s^{-2}$) value for smooth integration in the time domain and instead include all 3N-6 molecular vibrational modes in the summation. Thus, in our simulations, the linewidth at a given $T$ is explicitly obtained from the thermal population of the molecular normal modes, as described using a fully atomistic, quantum-mechanical, description of the vibrations. We proceed similarly for the simulation of the photoluminescence spectrum, replacing $\omega$ by $\omega^3$ for spontaneous emission and inverting the sign of $\omega_{ge}-\omega$ for emission instead of absorption.

## Data availability

The data that support the findings of this study are available from the corresponding authors upon request. Source data are provided as a Source Data file.

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

## Acknowledgements

This work has been financially supported by the FLAG-ERA Grant OPERA by DFG 437130745 and ANR-19-GRF1-0002-01, by the ANR-DFG NLE Grant GRANAO by DFG 431450789 and ANR-19-CE09-0031-01, by a public grant overseen by the French National Research Agency (ANR) as part of the "Investissements d'Avenir" program (Labex NanoSaclay, reference: ANR-10-LABX-0035). S.O. thanks the Polish National Science Center for funding (grant no. UMO-2020-39-I-ST4-01446). The computation was carried out with the support of the Interdisciplinary Center for Mathematical and Computational Modeling at the University of Warsaw (ICM UW) under grants no. G83-28 and GB80-24. The modeling activities in Mons were supported by the Belgian National Fund for Scientific Research (FRS-FNRS) within the Consortium des Équipements de Calcul Intensif—CÉCI, under Grant 2.5020.11, and by the Walloon Region (ZENOBE Tier-1 supercomputer, under grant 1117545). D.B. is FNRS Research Director. The authors acknowledge Dr. Elsa Cassette for helpful discussions.

## Author contributions

This work was conceived by J.-S.L. and S.C. The synthesis and characterization were performed by D.M.-L with help of G.H. S.C. and B.J. The photophysical characterization was performed by T.L. with the help of C.E., P.T., and L.R. The PLQY was determined by H.L.-F. DFT calculations were performed by S.O., N.R., and D.B. The manuscript was written by D.B., L.R., J.-S.L., and S.C. with input from all authors.

## Competing interests

The authors declare no competing interests.
