## [Peer Review File · Nature Communications]

nature portfolio

Peer Review FileReviewer comments, first round

Reviewer #1 (Remarks to the Author):

The manuscript by Medina-Lopez et al. presents a joint theoretical and experimental study on the synthesis and characterization of a new class of highly luminescent and well-defined graphene quantum dots. The paper is well-written and the message is clearly communicated. The results demonstrate the quality of the produced materials; their properties are convincingly rationalized with the combined experimental and first-principle analysis. The topic of this study is not brand new: graphene nanostructures have been a hot subject for research for more than 15 years now. Yet, the results shown in this work have some important points of novelty, especially in the chemical cleanliness of the synthesized structures. For these reasons, I believe that this manuscript can represent a suitable contribution to Nature Communications. However, before recommending publication, the authors should address the following points in order to prove the quality of their theoretical results and ensure their reproducibility.

- The information about the computational methods shared with the readers is rather scarce in the current version. I encourage the authors to include in the supporting information a section dedicated to theory. Specifically, the optimization of the systems in the excited states is not a standard procedure and it would be good if the authors could describe in detail the approach they used. Was the electronic structure constrained during optimization by adjusting the orbital occupations according to the contributions of the transition coefficients? What is the character of the excitations considered as starting point for the relaxation process? Are they dominated by a single occupied-virtual orbital transition or do many of those contribute? Is it possible to share these values with the readers? Also, the aforementioned constrained optimization often leads to numerical instabilities: did the authors encounter them in their calculations? If so, how did they overcome them? Can the authors provide more details regarding the calculations of the vibronic progression? Some equations would help ensure the reproducibility of the results.

- Ab initio calculations were performed using the HSE functional. This choice is nowadays standard for solids but in the context of molecules, other range-separation recipes such as the CAM-B3LYP are more popular. Can the authors discuss their choice for the exchange-correlation functional?

- Did the authors benchmark the accuracy of the adopted basis set?

- Which method was adopted to access vibrational frequencies? Were they calculated using density-functional perturbation theory or another approach?

- In the discussion of Fig. 1d, the authors mention two vibrations at 166 and 205 meV that couple with the $S_0 \rightarrow S_1$ transition. Can they provide more information about these modes? Which atoms/bonds are involved? As a side remark, adopting energy units instead of wave numbers is a bit unusual in the context of vibrations. Reporting both units or providing conversion tables in the Supporting Information would be helpful.

- In Fig. S6 several conformers of the same compounds are reported. How do these structures compare with each other energetically? Which one has the lowest energy?

- On page 4, just above Fig. 2, the authors write "no interconversion can be assumed". What is the basis for this statement?

- The maximum marked by a star in Fig. 1b is not clearly identifiable. I recommend the authors use an arrow pointing at it, instead.

- In Fig. 3a, what are units/scale of the x and y axes?

Reviewer #2 (Remarks to the Author):

In this paper, rod-like GQDs with good solubility and high PLQY were synthesized. The properties of GQDs were described through optical experiments and TDDFT calculations, and it was proposed that the prepared GQDs could be used in quantum emitter and sensing. However, there are still many problems in the characterization and mechanism analysis of GQDs. Firstly, the basic structure characterization of GQDs (HRTEM, XRD, Raman, XPS, etc.) is missing. Secondly, the optical characterization and analysis of GQDs are not clear enough. Finally, there is a lack of theoretical basis or literature support for some of the views in this paper. Therefore, publication of this article in Nature Communication is not recommended. The following are some suggestions:

1. Only MALDI-TOF MS characterization is not enough for the structure analysis of GQDs C96-tBu8, C78-tBu6, and C114-tBu10. More characterizations like ¹H NMR, ¹³C NMR and single-crystal structure should be provided.
2. More evidences like HRTEM, XRD, and Raman spectra should be provided to prove that the products belong to GQDs rather than organic molecules.
3. It is described that the rod-shaped GQDs could show high PLQY up to 94%. The result is interesting. The authors are suggested to provide the raw spectra (FLS980 fluorescence spectrofluorometer) to make the data more convincing.
4. Considering the rod-shaped GQDs are highly soluble, it is essential to provide the solubilities of the GQDs in various solvents.
5. Absorption and fluorescence spectra of GQDs at high concentration are required to demonstrate that the GQDs can remain relatively their individualization at high concentrations.
6. The authors are suggested to provide proof for the idea that the highly matched PLE and absorption spectra represent that C96-tBu8 are highly individualized in solution.
7. Fluorescence spectra of the GQDs under different excitation wavelengths are suggested to provide.
8. The PL of C96-tBu8 includes two peaks, please elucidate the reason of this phenomenon.
9. Please explain what the two fluorescence lifetimes represent in the TRPL spectra of C78-tbu6, and add their respective proportions.

Reviewer #3 (Remarks to the Author):

Graphene is an important low-dimensional quantum material. Graphene quantum dots (GQDs) are its molecular realizations holding a number of important technological applications. The present work reports synthesis and characterization of specific high purity GQDs showing high emission capacity due to lack of aggregation. Quantum-chemical simulations and various spectroscopic results help to establish structure-property relationships.

Overall, this study is very interesting and presents interesting materials systems with promising electronic properties. Given current topical interest in quantum and low-dimensional materials, this paper can potentially be published in Nature Communications. I hope that the authors will be able to address these issues in the revision.

1. Main: While the authors convincingly show the lack of aggregation of C96-tBu8 GQD that underpins its nice emissive capacity, there is no discussion on the underlying physical mechanisms (except statement that the tert-butyl groups ensure a good solubility). However, this argument fails in the case of C78tBu6 GQD which does exhibit aggregation. What suppresses dispersive interactions in the present case? Steric interactions introduced by tert-butyl? Is there any crossover in terms of prevalent interaction? Altogether, this is an important point that deserves further attention since any insights may provide useful molecular design guidelines for future studies.

2. An explanation of * peak in absorption in terms of conformer presence is not fully convincing. For example, "We can safely assume that there is no interconversion among the different conformers" – Why? What is a typical barrier height? This is relatively easy to evaluate from simulations. Further, ~16 meV (theory) vs 40 meV (experiment) difference seems to be on the higher side. Possible explanation in terms of "liquid or solid environment" is brought without any

proof. At least this requires some further simulations (see point below).

3. Modeling should comment on the solvent effects. Looks like all calculations were done in the gas-phase. It is easy to repeat some simulations within polarizable continuum model. Does solvent contribute to the Stokes shift (experimental value is 36 meV, which seems to be underestimated in simulations)?

4. Minor: Some rationale for the choice of HSE DFT functional should be spelled out. HSE was designed for periodic systems to address long-range effects. For the finite system some standard models like B3LYP or PBE0 or LR-corrected models would be more common. Further, I would expect that various ΔE discussed would be larger, for example, for LR-corrected models and to be closer to experimental values.

5. Minor: TDDFT transition densities perhaps require more explanation on what these quantities show for a broad audience.

6. Minor: 'Impulse response function (IRF)' in Fig. 1c acronym needs to be introduced in the caption.

7. Minor: "Stoke shift" -> Stokes shift

8. Minor: some language can be improved. For example, page 3 "In our seminal work" used in reference to the present paper. "to fight against" – perhaps 'to overcome the problem'

9. Minor: should structures on Fig. 4a be reverted, i.e. dendrimer -> GQD instead of GQD-> dendrimer?

Reviewer #4 (Remarks to the Author):

The present submission is focused on creating solution phase dispersions of single graphene quantum dots by functionalizing the quantum dot edges with tert-butyl groups that sterically suppress aggregation. They specifically focus on a set of planar structures that are fully aromatic (not characterized by the presence of localized double bonds). These structures yield maximally delocalized electronic states but present a large surface area that otherwise promotes aggregation. By realizing solution phase dispersions of individual quantum dots, they are then able to carry out ensemble studies of the fundamental optical properties of individual quantum dots than would otherwise be possible by, for example, single-molecule studies.

In two rod-shaped structures with 96 and 114 aromatic C atoms the authors have realized solution phase dispersions dominated by single quantum dots as seen in the facts that (1) the high reported photoluminescence quantum yield (PL QY) of 94%, (2) narrow absorption and emission spectra in solution, (3) solution phase PL spectra that are very close matches to PL spectra from single-molecule experiments, and (4) close correspondence between experimental spectral peak positions and polarization-resolved photoluminescence excitation spectra and PL and polarization-dependent absorption spectra. In contrast, for a rod-shaped 78 aromatic C atom structure and a triangular 96 aromatic C atom structure become more soluble when functionalized with tert-butyl groups, but they still exhibit aggregation; this is especially pronounced in the triangular structure. This can be easily understood by simply superimposing images of each type of quantum dot structure and rotating the images. There is a small range of angles for C78 and a substantially larger range of angles for triangular C96 over which the tert-butyl groups are not overlapping, but in rod-shaped C96 and C114 it is much more difficult to avoid such overlap.

The PL QYs for rod-shaped C96 and C114 are the highest PL QYs reported in graphene quantum dots of at least ~100 aromatic C atoms. Moreover, with each new graphene quantum dot or large polycyclic aromatic hydrocarbon (PAH) molecule, there is substantial chemical synthesis and characterization that must be done to realize new shapes or functionalization. However, the functionalization of PAH edges with tert-butyl groups has long been used to inhibit aggregation of large PAHs. Indeed, the realization of large PAHs (hexa-peri-hexabenzobis-peri-octacene or

HBPO, with 86 C atoms in the aromatic core) using tert-butyl groups to inhibit aggregation and realize high photoluminescence quantum yield (79% PL QY) in a very close energy range to the quantum dots studied here (the main low-energy emission peak of HBPO was at 602 nm instead of about 622 nm as seen here in rod-shaped C96) was demonstrated last year by Klaus Mullen's group (Gu et al., *Angew. Chem.* 61, e202201088 (2022)).

The authors have done a careful, thorough set of experiments. The main advances are that the optical properties of the solution phase dispersions are dominated by individual quantum dots, which enables detailed ensemble studies, and the PL QY is very high. The 94% and 91% PL QY for rod-shaped C96 and C114, respectively, are much higher than what they earlier estimated in a *Nature Communications* paper on single-dot studies of triangular C96 functionalized with simply alkyl chains, and in terms of certain applications, a 94% PL QY is a substantial improvement over the 79% seen in HBPO from Mullen's group. This is highly relevant for applications in a variety of fields. However, this is achieved by what is in many respects a minor variation on the widely used them of using functionalization (often with tert-butyl) to sterically hinder the aggregation of PAHs synthesized by Scholl oxidative cyclodehydrogenation. Combined with the fact that triangular C96 still aggregates when functionalized, this illustrates once again the general rule in the field that steric-hindrance-based approaches to inhibiting aggregation are very much shape-specific.

Response to the reviewers

Reviewer #1 (Remarks to the Author):

The manuscript by Medina-Lopez et al. presents a joint theoretical and experimental study on the synthesis and characterization of a new class of highly luminescent and well-defined graphene quantum dots. The paper is well-written and the message is clearly communicated. The results demonstrate the quality of the produced materials; their properties are convincingly rationalized with the combined experimental and first-principle analysis. The topic of this study is not brand new: graphene nanostructures have been a hot subject for research for more than 15 years now. Yet, the results shown in this work have some important points of novelty, especially in the chemical cleanliness of the synthesized structures. For these reasons, I believe that this manuscript can represent a suitable contribution to Nature Communications. However, before recommending publication, the authors should address the following points in order to prove the quality of their theoretical results and ensure their reproducibility.

Answer: We thank the reviewer for his/her positive evaluation of our work and for his/her constructive review that help us to improve our manuscript.

- The information about the computational methods shared with the readers is rather scarce in the current version. I encourage the authors to include in the supporting information a section dedicated to theory. Specifically, the optimization of the systems in the excited states is not a standard procedure and it would be good if the authors could describe in detail the approach they used. Was the electronic structure constrained during optimization by adjusting the orbital occupations according to the contributions of the transition coefficients?

Answer: From the comment by the reviewer, it seems she/he is more versed with solid-state electronic structure calculations. The approach used here to compute excited-state geometries of molecular systems *is in fact a very standard procedure*, at least when it deals with the lowest electronic excited states. It is based on the calculation of a force-constant matrix at the TDDFT level, as implemented in the Gaussian package. We refer the interested readers to the Gaussian manual where detailed information is provided on the methodology used and the variants available. To address the reviewer's question, we did not impose any constraint on the *orbital occupations*, but rather search for the equilibrium geometry *on the lowest singlet excited-state potential energy surface* in the framework of the Born-Oppenheimer approximation.

What is the character of the excitations considered as starting point for the relaxation process?

Answer: We do not impose any constraint on the character of the excitation used as starting point, except for the spin multiplicity (singlet). In the conjugated molecules investigated, the low-lying electronic excitations are all $\pi-\pi^*$ transitions.

Are they dominated by a single occupied-virtual orbital transition or do many of those contribute?

Answer: The lowest singlet excited state turns out to be dominated by a single transition involving the most frontier HOMO and LUMO orbitals, while higher-lying excited states have a somewhat more mixed character. The description of all relevant electronic excitations in terms of major electronic configurations is provided in Table S2.

Is it possible to share these values with the readers?

Answer: Table S2 collects absorption and emission wavelengths, oscillator strength as well as a molecular orbital description of the main electronic transitions for all molecules.

Also, the aforementioned constrained optimization often leads to numerical instabilities: did the authors encounter them in their calculations? If so, how did they overcome them?

Answer: No instabilities were observed during the optimization. Again, we did not impose any constraint apart from staying on the lowest singlet excited-state potential energy surface. As that state is energetically separated both from the ground state and higher-lying states in the Franck-Condon region, geometry optimization proceeds smoothly in all cases.

Can the authors provide more details regarding the calculations of the vibronic progression? Some equations would help ensure the reproducibility of the results.

Answer: We agree with the reviewer the approach used to model vibronic progression in the optical spectra should have been described in some more detail. We have now added a section on this in SI:

“To model the vibronic progression in the optical absorption and emission spectra, we start with the calculation of the exciton-phonon coupling constants and related Huang-Rhys factors, S_j , applying an undistorted (that is we assume the same normal modes in the ground, g , and lowest excited state, e , that means that we ignore Duschinsky matrix rotation effects) but displaced (or shifted) harmonic oscillator model. Thus, we project the displacements between the two optimized geometries in a Cartesian framework onto the ground-state (for absorption) or excited-state (for emission) normal modes:

$$d_j = Q_j(e) - Q_j(g)$$

The dimensionless Huang-Rhys factor for normal mode j is given by:

$$S_j = \frac{1}{2\hbar} \omega_j d_j^2$$

where $\omega_j / 2\pi$ is the corresponding normal-mode frequency. We then compute the optical absorption spectrum in the Franck-Condon approximation using:¹

$$\langle \alpha(\omega) \rangle = \omega \int_{-\infty}^{\infty} dt e^{it(\omega_{ge} - \omega) - \frac{D_{ge} t^2}{4} - \gamma_{ge} |t|} \exp \left[- \sum_j S_j \left\{ (2\bar{n}_j + 1) - (\bar{n}_j + 1) e^{it\omega_j} - \bar{n}_j e^{-it\omega_j} \right\} \right]$$

where ω_{ge} represents the adiabatic (0-0) transition energy from the ground state to the excited state,

$\bar{n}_j = (e^{\frac{\hbar\omega_j}{kT}} - 1)^{-1}$ is the Bose-Einstein population of the vibrational normal modes at temperature T ,

and $\gamma_{ge} (D_{ge})$ quantifies (in)homogeneous broadening. We set γ_{ge} to zero and D_{ge} to a very small (10^{-4} s^{-2}) value for smooth integration in the time domain and instead include all 3N-6 molecular vibrational modes in the summation. Thus, in our simulations, the linewidth at a given T is explicitly obtained from the thermal population of the molecular normal modes, as described using a fully atomistic, quantum-mechanical, description of the vibrations. We proceed in a similar fashion for the simulation of the photoluminescence spectrum, replacing ω by ω^3 for spontaneous emission and inverting the sign of $\omega_{ge} - \omega$ for emission instead of absorption.”

- *Ab initio* calculations were performed using the HSE functional. This choice is nowadays standard for solids but in the context of molecules, other range-separation recipes such as the CAM-B3LYP are more popular. Can the authors discuss their choice for the exchange-correlation functional?

Answer: HSE and range-separated functionals like CAM-B3LYP use different recipes to solve the self-interaction problem and asymptotic behavior of DFT functionals. The reviewer is completely right that, while HSE is now the reference for solids, it is not the case for molecules. However, in large π -conjugated systems such as those investigated here (and that are somehow in between small molecules and solids), we have previously demonstrated that HSE provides very good agreement between experimental and predicted excitation energies. This is the case for example in graphene nanoribbons, see L. Yang *et al.*, *Adv. Sci.* **9**, 2200708 (2022), X. Wang *et al.*, *J. Am. Chem. Soc.* **144**, 228-235 (2022) X. Yao *et al.*, *J. Am. Chem. Soc.* **143**, 5654-5658 (2021).

Nevertheless, to address the comment of the reviewer, we have performed additional TD-DFT excited-state calculations on the **C₇₈-tBu₆** GQD using three additional DFT functionals: CAM-B3LYP, PBE0 and wB97xD. The results displayed below (Fig. R1) show that, compared to HSE and the experimental results, CAM-B3LYP and wB97xD largely overestimate the excitation energies, with PBE0 doing a bit better. It is comforting to find that the overall shape of the optical absorption spectrum (namely the existence of two intense optical transitions) as well as the polarization of the electronic transitions and their nature is method independent. Yet, HSE unambiguously provides a more quantitative description. We now provide these additional results in SI (Fig. S8).

Fig. R1 TDDFT excited-state calculations on the **C₇₈-tBu₆** GQD in gas state using three additional DFT functionals: HSE, CAM-B3LYP, PBE0 and wB97xD.

- *Did the authors benchmark the accuracy of the adopted basis set?*

Answer: The basis set has been adopted based on previous (successful) attempts to model the electronic excitations in large polyaromatic hydrocarbons (see refs above: L. Yang *et al.*, *Adv. Sci.* **9**, 2200708 (2022), X. Wang *et al.*, *J. Am. Chem. Soc.* **144**, 228-235 (2022) X. Yao *et al.*, *J. Am. Chem. Soc.* **143**, 5654-5658 (2021)) and provides the right balance between accuracy and computational demand.

- *Which method was adopted to access vibrational frequencies? Were they calculated using density-functional perturbation theory or another approach?*

Answer: The full analytical hessian matrix was computed at the same level of theory as that used for geometry optimization.

- In the discussion of Fig. 1d, the authors mention two vibrations at 166 and 205 meV that couple with the S0 -> S1 transition. Can they provide more information about these modes? Which atoms/bonds are involved?

Answer: These are the two ubiquitous vibrational modes of polyaromatic hydrocarbons and involve collective stretching of the C-C and C=C bonds. We now provide visualization of these modes in SI (Fig. S10).

As a side remark, adopting energy units instead of wave numbers is a bit unusual in the context of vibrations. Reporting both units or providing conversion tables in the Supporting Information would be helpful.

Answer: Values in both meV and cm^{-1} have been added in the main text.

- In Fig. S6 several conformers of the same compounds are reported. How do these structures compare with each other energetically? Which one has the lowest energy?

Answer: Table S3 in SI reports the energy analysis of the different conformers. Considering the UD-DU as reference, UD-US has a comparable energy of 0.02 eV while all other conformers are less stable by 0.3-0.7 eV.

- On page 4, just above Fig. 2, the authors write "no interconversion can be assumed". What is the basis for this statement?

Answer: It is based on the high requested energy for converting one conformer into another, as reported in Table S3. In the revised manuscript we introduced a study on the aggregation and we showed that for $\text{C}_{96}\text{-tBu}_8$ two conformers (UD-DU and UD-UD) are present in majority in solution and the other are present at traces level (Table S6).

- The maximum marked by a star in Fig. 1b is not clearly identifiable. I recommend the authors use an arrow pointing at it, instead.

Answer: The change has been done following the referee's suggestion.

- In Fig. 3a, what are units/scale of the x and y axes?

Answer: The unit of the x and y axes in Fig. 3a is μm . The scale bar (10 μm) is given in white on the figure.

Reviewer #2 (Remarks to the Author):

In this paper, rod-like GQDs with good solubility and high PLQY were synthesized. The properties of GQDs were described through optical experiments and TDDFT calculations, and it was proposed that the prepared GQDs could be used in quantum emitter and sensing. However, there are still many problems in the characterization and mechanism analysis of GQDs. Firstly, the basic structure characterization of GQDs (HRTEM, XRD, Raman, XPS, etc.) is missing. Secondly, the optical characterization and analysis of GQDs are not clear enough. Finally, there is a lack of theoretical basis or literature support for some of the views in this paper. Therefore, publication of this article in Nature Communication is not recommended. The following are some suggestions:

1. Only MALDI-TOF MS characterization is not enough for the structure analysis of GQDs C96-tBu8, C78-tBu6, and C114-tBu10. More characterizations like 1H NMR, 13C NMR and single-crystal structure should be provided.

Answer: As suggested by the reviewer, we performed extensive NMR studies of the GQDs. We performed 1D NMR experiments (¹H NMR) and more importantly 2D NMR experiments (COSY and NOESY). Correlation spectroscopy (COSY) permits to identify the protons that are directly coupled in the molecules; it is the protons in red and in blue in the following figure. Nuclear Overhauser effect spectroscopy (NOESY) permits to identify the protons that are physically close to each other. With this experiment, we are able to correlate the protons from adjacent phenyl rings and protons close to *tert*-butyl groups. In conclusion, these experiments confirmed the structure of the GQDs. An example of spectra is given in Fig. R2 (below) for the C₉₆-tBu₈. These NMR experiments have been performed on all the GQDs (C₇₈-tBu₆, C₉₆-tBu₈, C₁₁₄-tBu₁₀ and the new C₁₃₂-tBu₁₂). We have added a sentence in the core of the article and we report all the spectra in Supplementary Information. The experiments have been performed by Dr. Gaspard Huber from the “Laboratoire Structure et Dynamique par Résonance Magnétique” (LSDRM) at CEA Paris-Saclay; we added him as co-author with the agreement of all the previous authors.

¹H NMR spectrum of C₉₆-tBu₈ (600 MHz, CS₂+THF-d₈ (250 μL + 350 μL), 298 K, [5.36·10⁻⁴ M])

¹H COSY spectrum

¹H NOESY spectrum

Fig. R2 Example of 1D and 2D ^1H NMR spectra obtained on the GQDs (here the $\text{C}_{96}\text{-tBu}_8$). The intense peaks at 2.21ppm and 3.58ppm are due to THF- d_8 .

Concerning ^{13}C NMR, despite the solubility of the GQDs, it was not possible to acquire exploitable 1D spectrum. However, we could obtain ^1H - ^{13}C HSQC (Heteronuclear Single Quantum Coherence) and ^1H - ^{13}C HMBC (Heteronuclear Multiple Bond Correlation) spectra (Fig. R3). HSQC experiment provides correlations between a carbon and its attached proton(s). HMBC correlates protons with carbons separated with two or three bonds. On this 2D spectrum, additional correlations appear between aromatic protons and carbon from *tert*-butyl groups. Those, we have reported as ^{13}C NMR characterization only the carbon detected *via* ^1H - ^{13}C HSQC and ^1H - ^{13}C HMBC in ESI.

^1H - ^{13}C HSQC spectrum

^1H - ^{13}C HMBC spectra

Fig. R3 Example of 2D ^1H - ^{13}C HSQC and HMBC NMR spectra obtained on the GQDs (here the C_{96} - $t\text{Bu}_8$).

Despite several attempts, we were not able to obtain reasonable single crystals of the GQDs. Indeed, it appeared that the GQDs formed amorphous or nanocrystalline precipitates that were not suitable for diffraction.

2. More evidences like HRTEM, XRD, and Raman spectra should be provided to prove that the products belong to GQDs rather than organic molecules.

Answer: First, the GQDs presented in this manuscript are *in fact* organic molecules exhibiting controlled compositions and structures. Due to their size and number of carbon atoms they are called graphene quantum dot or nanographene by the community. Here we do not understand the referee's remark and strongly disagree with his/her statement. Even if any addition of data is always valuable, we think that all the characterizations provided in the new version of the manuscript are more than sufficient to prove the final structure of final products. While we agree that single-crystal X-ray diffraction would bring valuable information, XRD and Raman are not relevant here.

Concerning HRTEM, the remark of the reviewer is pertinent and actually we tried to performed HRTEM on our GQDs (in particular C_{96} - $t\text{Bu}_8$) deposited on graphene. Unfortunately, the results were not conclusive conversely to our previous study on the triangular-shaped C_{96} - $(\text{C}_{12}\text{H}_{25})_6$ GQDs (see the ESI of T. Liu *et al. Nanoscale* **14**, 3826 (2022)). The triangular-shaped C_{96} - $(\text{C}_{12}\text{H}_{25})_6$ GQDs tend to form columnar aggregates that are easily discernable on the TEM grid. The rod-shaped GQDs reported here are designed to not form aggregates and they are extremely difficult to distinguish on the TEM grids from contamination coming from the solvent or from the preparation of the graphene-coated grid.

3. It is described that the rod-shaped GQDs could show high PLQY up to 94%. The result is interesting. The authors are suggested to provide the raw spectra (FLS980 fluorescence spectrofluorometer) to make the data more convincing.

Answer: Following referee's suggestion, we add all the measurements in the supporting information, including the one of Fluorescein that serves as calibration tool

4. *Considering the rod-shaped GQDs are highly soluble, it is essential to provide the solubilities of the GQDs in various solvents.*

Answer: The remark of the reviewer is correct and we think that the term “highly soluble” can be misleading. In this manuscript, the term “highly soluble” refers to the complete individualization of the GQDs in solution. Therefore, we replace the terms “soluble” by “individualized” in the manuscript. By optical absorption, photoluminescence and photoluminescence spectroscopy, we proved that the GQDs are perfectly individualized in 1,2,4-trichlorobenzene (excepted the smallest one: **C₇₈-tBu₆**) and we show also in the new version of Fig. S5 that the absorption spectra of **C₉₆-tBu₈**, **C₁₁₄-tBu₁₀** and **C₁₃₂-tBu₁₂** remained identical in the range of 10^{-7} to 10^{-5} M. These concentrations are the typical concentration in which these molecules are used. It is even lower for single molecule spectroscopy, for which typically nM solutions are prepared. In this revised version, we also provide the absorption and PL spectra of the GQDs in various solvents (dichloromethane, tetrahydrofuran and toluene). The spectra absorption and PL spectra are very similar (once again, expected for **C₇₈-tBu₆**) in all solvents considered (Fig. S12) although the best solvent remains 1,2,4-trichlorobenzene.

Finally, it is worth mentioning that the GQDs present a good dispersibility at higher concentration ($\sim 10^{-4}$ M) used for NMR experiments. We did not observe by eye aggregates or precipitates in the NMR tubes after 2 days of experiments.

5. *Absorption and fluorescence spectra of GQDs at high concentration are required to demonstrate that the GQDs can remain relatively their individualization at high concentrations.*

Answer: The referee is right to say that the notion of solubility must be defined in relation to the concentration at which one works. First, it has been well reported that, even at low concentration, such large objects were not fully individualized in solution even though they are “soluble” (see for instance the works of K. Müllen,^{2,3} Y.-Z. Tan^{4,5} or our work⁶). Therefore, reaching the complete individualization even at a low concentration is an important result. Secondly, the concept of high concentration is relative. In this study, the nominal concentration is of the order of few 10^{-5} M. At these concentrations, no indication of aggregation has been observed. For the applications targeted by our consortium, this is a high concentration. For the single molecule measurements, we have to dilute the mother solution by factors from 100 to 1000.

6. *The authors are suggested to provide proof for the idea that the highly matched PLE and absorption spectra represent that C96-tBu8 are highly individualized in solution.*

Answer: Photoluminescence excitation spectroscopy is a powerful tool that allows to connect the quantum states at the origin of the emission to the ones that absorb light. Therefore, PLE will probe only emissive species. In order to understand our statement, one has first to keep in mind that large and disordered aggregates present in general a very low fluorescence quantum yield. Therefore, one does not see them in the fluorescence experiments when some monomers with high quantum yield are present. Secondly, when a suspension is mainly composed of aggregates, the absorption spectrum is broadened. This is because absorption spectroscopy probes all the objects in the solution. This is for instance the case for the triangular-shaped **C₉₆-(C₁₂H₂₅)₆** as described in Fig. R4 from the SI of our 2018 paper.⁶ In this example, the suspension contains only a little amount of individualized GQDs. In contrast with the absorption spectrum, the PLE shows much more defined lines. This is because the PLE spectrum indeed corresponds to the absorption of the individualized GQDs. On the contrary, the absorption spectrum of the new family of GQDs reported here shows well defined lines. This is a first strong indication that the GQDs are well individualized. PLE spectrum probing the individualized GQDs, the fact that it matches perfectly the absorption spectrum reinforces the conclusion of a high individualization of these GQDs.

We modified the sentence in the paper to make it clearer.

Fig. R4 Comparison between the absorption (in blue) and PLE (in green and red) spectra of triangular-shaped $C_{96}-(C_{12}H_{25})_6$ in 1,2,4-trichlorobenzene; extracted from ESI of reference 2.

7. Fluorescence spectra of the GQDs under different excitation wavelengths are suggested to provide.

Answer: Fluorescence spectra at two excitation wavelengths have been provided in the supplementary information (Fig. S14). One can observe that the shape of the spectrum is not influenced by the excitation wavelength

8. The PL of C_{96} -tBu8 includes two peaks, please elucidate the reason of this phenomenon.

Answer: The main line in the fluorescence spectrum is known as the zero-phonon line of the emission. The second one at ~ 170 meV is a vibronic line corresponding to the C=C stretching mode. We add a sentence in the paper to make it clearer.

9. Please explain what the two fluorescence lifetimes represent in the TRPL spectra of C_{78} -tBu6, and add their respective proportions.

Answer: As mentioned in the text, the C_{78} -tBu6 is a special case since we do observe monomers and dimers in the solution. The two lifetimes probably arise from the two different species. Since the two species should have different quantum yield, the proportion between both exponential is not a relevant parameter. We also performed calculations that explain the tendency of C_{78} -tBu6 to form aggregate and we show that it does not happen with C_{96} -tBu8 and larger GQDs.

Reviewer #3 (Remarks to the Author):

Graphene is an important low-dimensional quantum material. Graphene quantum dots (GQDs) are its molecular realizations holding a number of important technological applications. The present work reports synthesis and characterization of specific high purity GQDs showing high emission capacity due to lack of aggregation. Quantum-chemical simulations and various spectroscopic results help to establish structure-property relationships.

Overall, this study is very interesting and presents interesting materials systems with promising electronic properties. Given current topical interest in quantum and low-dimensional materials, this

paper can potentially be published in Nature Communications. I hope that the authors will be able to address these issues in the revision.

Answer: We thank the referee for his/her positive statement on our work.

1. *Main: While the authors convincingly show the lack of aggregation of C₉₆-tBu₈ GQD that underpins its nice emissive capacity, there is no discussion on the underlying physical mechanisms (except statement that the tert-butyl groups ensure a good solubility). However, this argument fails in the case of C₇₈tBu₆ GQD which does exhibit aggregation. What suppresses dispersive interactions in the present case? Steric interactions introduced by tert-butyl? Is there any crossover in terms of prevalent interaction? Altogether, this is an important point that deserves further attention since any insights may provide useful molecular design guidelines for future studies.*

Answer: We performed calculations of the Potential of Mean Force (PMF) curves corresponding to C₇₈-tBu₆ and C₉₆-tBu₈ dimer formation, using Umbrella Sampling (US) and the Weighted Histogram Analysis Method (WHAM)⁷ (Fig. 5). Model systems contain two GQDs in their most stable conformation (2 C₇₈-tBu₆ U-U or 2 C₉₆-tBu₈ UD-DU) solvated in a cubic box of 1,2,4-trichlorobenzene molecules. The picture that emerges from the simulations is the following. As long as the molecules can adopt a conformation with all (most) tert-butyl groups pointing outward, aggregation may proceed through the sliding of one molecule on top of the other. This situation can happen for any couple of C₇₈-tBu₆ conformers: considering any combination of U-U and U-D, there is always a chance to have a dimer side with all tert-butyl pointing outward. On the contrary, this is not possible in the two most stable conformers of C₉₆-tBu₈, UD-DU and UD-UD, since a tert-butyl pointing outward is always seating next to a tert-butyl pointing inward on the same side of the molecule. While such an aggregation pathway would, in principle, be possible for the other conformers (with adjacent U or D side groups), these are much less stable as they involve much larger intramolecular strain and very likely not present in solution. In conclusion, we found that aggregation can be avoided in GQDs by adding tert-butyl on the molecules if the most stable conformers show adjacent tert-butyl groups pointing in opposite directions, as turns out to be the case in C₉₆-tBu₈ (and higher GQDs) but not in C₇₈-tBu₆.

A new figure and the full description has been added in the last part of the manuscript entitled “**GQD Aggregation**”.

2. *An explanation of * peak in absorption in terms of conformer presence is not fully convincing. For example, “We can safely assume that there is no interconversion among the different conformers” – Why? What is a typical barrier height? This is relatively easy to evaluate from simulations. Further, ~16 meV (theory) vs 40 meV (experiment) difference seems to be on the higher side. Possible explanation in terms of “liquid or solid environment” is brought without any proof. At least this requires some further simulations (see point below).*

Answer: The relative energy of the different conformers is reported in SI, for both gas phase and solvent. Considering the UD-UD as reference, we observe that only the UD-DU conformer has similar energy, lower of 0.02 eV in both gas phase and solvent (Table S3). In addition, the energy of the remaining conformers is much higher, in the 0.30-0.71 eV range in both gas phase and solvent environments. This already justifies our assertion that “there is no interconversion among the different conformers” since the process is thermodynamically unfavorable.

The difference in gas phase excitation energy reported in the manuscript should be 20 meV, not 16. This has been corrected in the revised version of the manuscript. The solvent has no effect on the difference in excitation energy for the different conformers, as the maximum difference remains 20

meV (data added in SI, Table S3). Instead, the presence of the solvent redshifts the absorption spectra of ca. 15 nm for all the conformers.

3. *Modeling should comment on the solvent effects. Looks like all calculations were done in the gas-phase. It is easy to repeat some simulations within polarizable continuum model. Does solvent contribute to the Stokes shift (experimental value is 36 meV, which seems to be underestimated in simulations)?*

Answer: Indeed, all calculations have been performed in the gas phase. We are not expecting large solvatochromic shifts in PAHs. Nevertheless, along the line suggested by the reviewer, we have repeated the HSE calculations for **C₇₈-tBu₆**, **C₉₆-tBu₈** and **C₁₁₄-tBu₁₀** using PCM parameters for 1,2,4-trichlorobenzene, this is described in Fig. S8 and shown below (Fig. R5). As expected, there is a small (10-18 nm) red shift in the lowest optical absorption going from the gas to the solvent phase, which likely contributes to the measured Stokes shift. However, a quantitative assessment would require running excited-state calculations in the solvent and, possibly, going beyond an implicit continuous model for the solvent. These are lengthy and tedious calculations that go beyond the scope of this work.

Fig. R5 TDDFT calculations on the **C₇₈-tBu₆**, **C₉₆-tBu₈** and **C₁₁₄-tBu₁₀** GQD in 1,2,4-trichlorobenzene.

4. *Minor: Some rationale for the choice of HSE DFT functional should be spelled out. HSE was designed for periodic systems to address long-range effects. For the finite system some standard models like B3LYP or PBE0 or LR-corrected models would be more common. Further, I would expect that various Delta E discussed would be larger, for example, for LR-corrected models and to be closer to experimental values.*

Answer: The comment is similar to the one of reviewer 1. See above for the reply

5. *Minor: TDDFT transition densities perhaps require more explanation on what these quantities show for a broad audience.*

Answer: The remark of the reviewer is correct. We have added explanation in the main text.

6. *Minor: 'Impulse response function(IRF)' in Fig. 1c acronym needs to be introduced in the caption.*

Answer: The correction has been made.

7. *Minor: "Stoke shift" -> Stokes shift*

Answer: The correction has been made.

8. Minor: some language can be improved. For example, page 3 “In our seminal work” used in reference to the present paper. “to fight against” – perhaps ‘to overcome the problem’

Answer: We modified the sentence as following:

“In our previous work, we had to overcome the high aggregation tendency of the triangular-shaped GQDs bearing dodecyl alkyl chains ($C_{96}-(C_{12}H_{25})_6$) to perform single molecule spectroscopy.”

9. Minor: should structures on Fig. 4a be reverted, i.e. dendrimer -> GQD instead of GQD-> dendrimer?

Answer: Actually, the notation corresponds to the retrosynthetic representation of the synthesis of the GQDs. The arrows are retrosynthetic arrows used classically in chemistry to represent a structure and its precursor.

Reviewer #4 (Remarks to the Author):

The present submission is focused on creating solution phase dispersions of single graphene quantum dots by functionalizing the quantum dot edges with tert-butyl groups that sterically suppress aggregation. They specifically focus on a set of planar structures that are fully aromatic (not characterized by the presence of localized double bonds). These structures yield maximally delocalized electronic states but present a large surface area that otherwise promotes aggregation. By realizing solution phase dispersions of individual quantum dots, they are then able to carry out ensemble studies of the fundamental optical properties of individual quantum dots than would otherwise be possible by, for example, single-molecule studies.

In two rod-shaped structures with 96 and 114 aromatic C atoms the authors have realized solution phase dispersions dominated by single quantum dots as seen in the facts that (1) the high reported photoluminescence quantum yield (PL QY) of 94%, (2) narrow absorption and emission spectra in solution, (3) solution phase PL spectra that are very close matches to PL spectra from single-molecule experiments, and (4) close correspondence between experimental spectral peak positions and polarization-resolved photoluminescence excitation spectra and PL and polarization-dependent absorption spectra. In contrast, for a rod-shaped 78 aromatic C atom structure and a triangular 96 aromatic C atom structure become more soluble when functionalized with tert-butyl groups, but they still exhibit aggregation; this is especially pronounced in the triangular structure. This can be easily understood by simply superimposing images of each type of quantum dot structure and rotating the images. There is a small range of angles for C78 and a substantially larger range of angles for triangular C96 over which the tert-butyl groups are not overlapping, but in rod-shaped C96 and C114 it is much more difficult to avoid such overlap.

Answer: This comment is very similar to the one of reviewer 3 (Question 1). As mentioned the formation of dimers can be only observed for C_{78} -tBu₆. The comment is fully addressed in the last part of the manuscript entitled “GQD Aggregation”.

The PL QYs for rod-shaped C96 and C114 are the highest PL QYs reported in graphene quantum dots of at least ~100 aromatic C atoms. Moreover, with each new graphene quantum dot or large polycyclic aromatic hydrocarbon (PAH) molecule, there is substantial chemical synthesis and characterization that must be done to realize new shapes or functionalization. However, the functionalization of PAH edges with tert-butyl groups has long been used to inhibit aggregation of large PAHs. Indeed, the realization of large PAHs (hexa-peri-hexabenzobis-peri-octacene or HBPO, with 86 C atoms in the

aromatic core) using *tert*-butyl groups to inhibit aggregation and realize high photoluminescence quantum yield (79% PL QY) in a very close energy range to the quantum dots studied here (the main low-energy emission peak of HBPO was at 602 nm instead of about 622 nm as seen here in rod-shaped C96) was demonstrated last year by Klaus Mullen's group (Gu et al., *Angew. Chem.* 61, e202201088 (2022)).

The authors have done a careful, thorough set of experiments. The main advances are that the optical properties of the solution phase dispersions are dominated by individual quantum dots, which enables detailed ensemble studies, and the PL QY is very high. The 94% and 91% PL QY for rod-shaped C96 and C114, respectively, are much higher than what they earlier estimated in a *Nature Communications* paper on single-dot studies of triangular C96 functionalized with simply alkyl chains, and in terms of certain applications, a 94% PL QY is a substantial improvement over the 79% seen in HBPO from Mullen's group. This is highly relevant for applications in a variety of fields. However, this is achieved by what is in many respects a minor variation on the widely used theme of using functionalization (often with *tert*-butyl) to sterically hinder the aggregation of PAHs synthesized by Scholl oxidative cyclodehydrogenation. Combined with the fact that triangular C96 still aggregates when functionalized, this illustrates once again the general rule in the field that steric-hindrance-based approaches to inhibiting aggregation are very much shape-specific.

Answer: The remarks of the reviewer are correct. The idea of using *tert*-butyl groups is not new and has led to beautiful results in terms of GQD structures and properties. It is worth mentioning that in many cases the presence of *tert*-butyl groups on two different phenyls pointing toward the same direction was achieved to avoid the formation of the bond between the two phenyls giving rise to warped structures.

We agree with the reviewer the major results of this paper are the optical properties of the GQDs which have been made possible thanks to the individualization of the emitters in solution. The particular positioning of the *t*Bu groups along the GQDs has permitted this. It is true that it is shape-specific as shown in the present paper. Nevertheless, we think that, among other reasons, the novelty our work is based on the synthesis of a family of GQDs for which this approach is successful. In order to emphasize this, we decided to perform the synthesis of a fourth rod-shaped GQDs, namely the **C₁₃₂-*t*Bu₁₂** GQD. We add the results in the new version of the manuscript. In particular, it pushes the absorption to 687 nm with still a high quantum yield (88%) Moreover, the length of the GQDs could be further expand. Based on the same principle, we are synthesizing now larger GQDs containing 4 benzene rings in width (Fig. R6). The preliminary molecular mechanical calculations suggest that the structure can be sufficiently affected by the presence of the *t*Bu to be again individualized in solution.

Fig. R6 Example of larger GQDs envisioned now.

Additional comments:

In the revised version of the manuscript, we have also added a new GQD: **C₁₃₂-*t*Bu₁₂** (the next member of the rod-shaped GQD family) to show that our iterative synthesis can be used to further extend the size of the GQDs. While answering the questions of the reviewers, we found a mistake in the treatment

of our data concerning Fig. 4d. We checked it carefully and modify the figure accordingly. The conclusion is exactly the same as the one mentioned in the first version of the paper.

References

1. W. W. Parson, *modern optical spectroscopy*, Springer-Verlag Berlin Heidelberg (2007).
2. Debije, M. G., Piris, J., de Haas, M. P., Warman, J. M., Tomovic, Z., Simpson, C. D., Watson, M. D. & Müllen, K. The Optical and Charge Transport Properties of Discotic Materials with Large Aromatic Hydrocarbon Cores. *J. Am. Chem. Soc.* **126**, 4641-4645 (2004).
3. Wasserfallen, D., Kastler, M., Pisula, W., Hofer, W. A., Fogel, Y., Wang, Z. & Müllen, K. Suppressing Aggregation in a Large Polycyclic Aromatic Hydrocarbon. *J. Am. Chem. Soc.* **128**, 1334-1339 (2006).
4. Zhao, X.-J., Hou, H., Fan, X.-T., Wang, Y., Liu, Y.-M., Tang, C., Liu, S.-H., Ding, P.-P., Cheng, J., Lin, D.-H., Wang, C., Yang, Y. & Tan, Y.-Z. Molecular bilayer graphene. *Nat. Commun.* **10**, 3057 (2019).
5. Zhao, X.-J., Hou, H., Ding, P.-P., Deng, Z.-Y., Su, Y.-Y., Liu, S.-H., Liu, Y.-M., Tang, C., Feng, L.-B. & Tan, Y.-Z. Molecular defect-containing bilayer graphene exhibiting brightened luminescence. *Sci. Adv.* **6**, eaay8541 (2020).
6. Zhao, S., Lavie, J., Rondin, L., Orcin-Chaix, L., Diederichs, C., Roussignol, P., Chassagneux, Y., Voisin, C., Müllen, K., Narita, A., Campidelli, S. & Lauret, J.-S. Single photon emission from graphene quantum dots at room temperature. *Nat. Commun.* **9**, 3470 (2018).
7. Kumar, S., Bouzida, D., Swendsen, R. H., Kollman, P. & Rosenberg, J. M. The Weighted Histogram Analysis Method for Free-Energy Calculations on Biomolecules. I. The Method. *J. Comput. Chem.* **13**, 1011-1021 (1992).

Reviewer comments, second round

Reviewer #1 (Remarks to the Author):

I am satisfied with the authors' replies to my comments and questions, and to the corresponding revisions they made. I can now recommend publication.

Reviewer #2 (Remarks to the Author):

The quality of the revised manuscript has been greatly improved and I recommend accept it.

Reviewer #3 (Remarks to the Author):

The authors carefully considered the Referee's comments and appropriately modified the MS. I now recommend the article for publication in Nature Comm.

Reviewer #4 (Remarks to the Author):

The main question I had about the previous version of the manuscript largely remains. Is this work truly novel? The paper describes the synthesis of a class of graphene quantum dots designed to exhibit minimal aggregation at the concentrations needed for certain applications. Furthermore, the paper demonstrates that the dots exhibit minimal aggregation at the concentrations of interest and spectroscopic characterization of their optical properties, especially their excellent emissive properties (near unity photoluminescence quantum yields). As I stated in my earlier review and the authors acknowledge, the basic features of the synthesis are not new. Nonetheless, considering the synthesis, the demonstration of near unity quantum yields, and the entirety of the characterization (chemical, optical, and theoretical), I believe that the manuscript merits publication in Nature Communications.

I would like to highlight the quality and extent of the authors' characterization. The other referees raised some legitimate questions about theory and characterization, but the authors have gone to exceptional lengths to enhance the confidence the reader can place in the conclusions. This paper is more thorough than many highly cited papers in the field, which certainly adds to the impact that it may have.